

# A global marine particle size distribution dataset obtained with the Underwater Vision Profiler 5

Rainer Kiko[1,*], Marc Picheral[1,*], David Antoine[2,1], Marcel Babin[3], Léo Berline[4], Tristan Biard[5], Emmanuel Boss[6], Peter Brandt[7,8], Francois Carlotti[4], Svenja Christiansen[9], Laurent Coppola[1], Leandro de la Cruz[10], Emilie Diamond-Riquier[1], Xavier Durrieu de Madron[11], Amanda Elineau[12], Gabriel Gorsky[1], Lionel Guidi[1], Helena Hauss[7], Jean-Olivier Irisson[1], Lee Karp-Boss[6], Johannes Karstensen[7], Dong-gyun Kim[13], Rachel M. Lekanoff[14], Fabien Lombard[1], Rubens M. Lopes[10], Claudie Marec[3], Andrew M.P. McDonnell[14], Daniela Niemeyer[7], Margaux Noyon[15], Stephanie H. O'Daly[14], Mark Ohman[16], Jessica L. Pretty[14], Andreas Rogge[17,13], Sarah Searson[18], Masashi Shibata[19], Yuji Tanaka[20], Toste Tanhua[7], Jan Taucher[7], Emilia Trudnowska[21], Jessica S. Turner[22], Anya Waite[23], and Lars Stemmann[1]

[1]Sorbonne Université, CNRS, UMR 7093, Laboratoire d'Océanographie de Villefranche-sur-Mer (LOV), Villefranche-sur-Mer, France
[2]Remote Sensing and Satellite Research Group, School of Earth and Planetary Sciences, Curtin University, Perth, WA 6845, Australia
[3]Département de Biologie, Université Laval, Québec, Canada
[4]MIO, Aix Marseille Univ., Université de Toulon, CNRS, IRD, MIO, Marseille, France UMR 7294 Mediterranean Institute of Oceanography, Marseille, France
[5]Univ. Littoral Côte d'Opale, Univ. Lille, CNRS, UMR 8187 - LOG - Laboratoire d'Océanologie et de Géosciences, F-62930 Wimereux, France
[6]School of Marine sciences, University of Maine, Orono, Massachusetts, USA
[7]GEOMAR Helmholtz Centre for Ocean Research Kiel, Kiel, Germany
[8]Faculty of Mathematics and Natural Sciences, Kiel University, Kiel, Germany
[9]Department of Biosciences, University of Oslo, Oslo, Norway
[10]University of São Paulo, Oceanographic Institute, São Paulo, Brazil
[11]CEFREM, CNRS-Université de Perpignan Via Domitia, Perpignan, France
[12]Institut de la Mer de Villefranche, CNRS - Sorbonne Université, Villefranche-sur-Mer, FRANCE
[13]Alfred-Wegener-Institut Helmholtz-Zentrum für Polar- und Meeresforschung, Bremerhaven, Germany
[14]College of Fisheries and Ocean Sciences, University of Alaska Fairbanks, Fairbanks, Alaska, USA
[15]Nelson Mandela University, Institute for Coastal and Marine Research, Gqeberha, South Africa
[16]Scripps Institution of Oceanography, La Jolla, California, USA
[17]Institute for Ecosystem Research, Kiel University, Kiel, Germany
[18]National Institute of Water and Atmosphere Research, New Zealand
[19]SeaBreath Co., Ltd., Tokyo, Japan
[20]Tokyo University of Marine Science and Technology, Tokyo, Japan
[21]Department of Marine Ecology, Institute of Oceanology Polish Academy of Sciences, Sopot, Poland
[22]Department of Marine Sciences, University of Connecticut Avery Point, Groton, Connecticut, USA
[23]Dalhousie University, Halifax, Nova Scotia, Canada
[*]These authors contributed equally to this work.

**Correspondence:** Rainer Kiko (rainer.kiko@obs-vlfr.fr)





**Abstract.** Marine particles of different nature are found throughout the global Ocean. The term "marine particles" describes detritus aggregates, fecal pellets, but also bacterio-, phyto-, zooplankton and nekton. Here we present a global particle size distribution dataset obtained with several Underwater Vision Profiler 5 (UVP5) camera systems. Overall, within the 64 $\mu$m to about 50 mm size range covered by the UVP5, detrital particles are the most abundant component of all marine particles in this size range and thus measurements of the particle size distribution with the UVP5 can yield important information on detrital particle dynamics. During deployment, which is possible down to 6000 m depth, the UVP5 images a volume of about 1 L at a frequency of 6 to 20 Hz. Each image is segmented in real time and size measurements of particles are automatically stored. All UVP5 units used to generate the here presented dataset were inter-calibrated using a UVP5 High Definition unit as reference. Our consistent particle size distribution dataset contains 8805 vertical profiles collected between 2008-06-19 and 2020-11-23. All major ocean basins, as well as the Mediterranean and the Baltic Sea were sampled. 19% of all profiles had a maximum sampling depth shallower than 200 dbar, 80% had a maximum sampling depth greater than 200 dbar, 38% sampled at least the upper 1000 dbar depth range and 11% went down to at least 3000 dbar depth. First analysis of the particle size distribution dataset shows that particle abundance is found to be high at high latitudes and in coastal areas where surface productivity or continental inputs are elevated. Lowest values are found in the deeep ocean and in the oceanic gyres. Our dataset should be valuable for more in-depth studies that focus on the analysis of regional, temporal and global patterns of particle size distribution and flux as well as for the development and adjustment of regional and global biogeochemical models. The marine particle size distribution dataset (Kiko et al., 2021) is available at https://doi.pangaea.de/10.1594/PANGAEA.924375.

# 1 Introduction

## 1.1 Nature and origin of marine particles

Bacterio-, phyto- and zooplankton, nekton, aggregates, marine snow, fecal pellets, biomineralized shells, mineral dust, precipitates, suspended clay and nowadays also plastics are part of the general marine particle size spectrum (Sheldon and Parsons, 1967; Stemmann and Boss, 2012; Cózar et al., 2014). The relative contribution of living, detrital and abiotic particles to the total load of particles is not well known and may vary according to the particle size range and the marine ecosystem investigated from 1% to 50% (Forest et al., 2012; Stemmann and Boss, 2012; Checkley Jr et al., 2008). Abiotic particles can originate from resuspension at the seabed (Puig et al., 2013; McCave, 2009, 1986; Honjo et al., 1984), dust deposition (Zuniga et al., 2008; Ratmeyer et al., 1999) and influx by rivers (Ludwig and Probst, 1998) and glaciers (Neal et al., 2010). Furthermore, dissolved constituents precipitate when riverwater (Many et al., 2019) or hydrothermal fluids (German and Von Damm, 2003) mix with seawater. Photosynthesis by planktonic algae is the almost exclusive source of biogenic carbon in the open ocean, although other processes such as carbon fixation by chemoautotrophs (e.g., at hydrothermal vents), benthic algae, seagrass and, moreover, land and river derived organic particles add to this as well (e.g., Duarte et al. (2010); Ludwig and Probst (1998)). Higher trophic levels consume this biogenic carbon to build up biomass and fuel physiological activity. Along the entire plankton trophic web, part of the consumed carbon is also transformed into detritus (fecal pellets, exuviae, discarded houses or dead bodies). Small particles such as phytoplankton cells can also coagulate to form larger aggregates, that might also include other





detrital particles (Jackson, 1990). The two pathways lead to the formation of detrital particles, with different sinking properties
depending on their size, content and porosity (Stemmann et al., 2004). As particle size is an essential trait for many biotic and
abiotic interactions, it is often used to develop and calibrate size resolved mechanistic models of phytoplankton bloom for-
mation, particle coagulation and export to the mesopelagic zone (Stemmann et al., 2004; Jouandet et al., 2014; Bianchi et al.,
2018). Moreover, the size structure of particles and plankton is one of the most relevant indicators of ecosystem functionality
and energy fluxes (Jackson, 1990; Zhou; Stemmann and Boss, 2012). How abiotic and biotic marine particles of different sizes
are formed, destroyed, advected or sink are key questions in ocean carbon cycling and biogeochemistry (Stemmann et al.,
2012; Boyd et al.; Giering et al., 2020) and therefore their quantitative monitoring is needed.

## 1.2 Marine particle imaging

Many phyto- and zooplankton organisms, but also some other particles are sturdy and can be sampled using nets, traps, sedi-
ment traps, bottles and in situ filtration devices. Fragile particles often formed by aggregation of diverse source particles (dead
cells, fecal pellets, exudates, minerals) called "Marine snow" (Beebe, 1931) and fragile zooplankton such as cnidarians, rhizar-
ians and other gelatinous organisms are however not amenable to such sampling methods (O'Hern et al.; Alldredge and Silver,
1988; Wiebe and Benfield, 2003; Remsen et al., 2004). Therefore, only in situ measurements allow for a realistic assessment of
the size and abundance of marine particles (Alldredge and Silver, 1988). Earliest such measurements were made from moored
platforms, submersibles or by divers and included analyses of photographic images (Suzuki and Kato, 1953; Alldredge and
50 Gotschalk, 1988). Advancement in electronic components and digital processing routines then allowed for the development
of instruments such as the optical plankton counter (Herman, 1992), holographic instruments Katz et al. (1999) and various
camera systems Asper (1987); Honjo et al. (1984); Lampitt et al. (1993); Ratmeyer and Wefer (1996); Benfield et al. (2007).
Among them the Underwater Vision Profilers (UVP) Gorsky et al. (2000); Picheral et al. (2010) was designed to automatically
size and count undisturbed abiotic and biotic marine particles.

## 1.3 The Underwater Vision Profiler and its use

The UVP was developed at the Laboratoire d'Océanographie de Villefranche (LOV) to provide consistent measurements of
particle abundance and size. Single units of UVP versions 1 to 4 were produced from 1991 to 2008 (Gorsky et al., 2000,
2002). The first prototype (sn000) of version 5 started field operations in 2008 and was described in detail by Picheral et al.
(2010). The instrument was commercialised in 2010 and produced until 2021. A standard (STD) version with a 1.3 Megapixel
greyscale camera was produced between 2008 and 2016 (serial numbers 000 to 011) and a high-definition (HD) version with
a 4 Megapixel greyscale camera was produced between 2016 and 2021 (serial numbers 200 to 223). The smaller and more
versatile UVP6 (Picheral et al., 2022) is commercially available since 2019. In the standard setting, the UVP5 images a volume
of about 1 L at a frequency of 6 to 20 Hz and can be deployed down to 6000 m depth. Particles on each image are automatically
sized. Further data processing of all particles allows calculations of the particle size distribution - the particle abundance or
biovolume in increasing size intervals. The UVP5 STD version covers the size range from 102 $\mu$m to $\sim$ 50 mm ESD, the HD





version the size range from 64 $\mu$m to $\sim$ 50 mm ESD. Through reduction of the distance between the LED lights and the camera, the resolution can be further increased, but then the imaged volume is reduced. Inter-calibrated UVP5 units are globally in use by several teams. Since the UVP5 is mostly integrated in the CTD-Rosette and has its own pressure sensor, fine-scale vertical

distribution of particles and major plankton groups can be related to environmental data obtained with other sensors mounted on the CTD-rosette. Most efforts regarding the analyses of UVP particle size spectra (including data from earlier versions such as the UVP4) have been on the estimation of particle biomass and flux by comparing them with particulate organic carbon (POC) collected in sediment traps or Niskin bottles (e.g., (Guidi et al., 2008b, 2015; Kiko et al., 2017; Stemmann et al., 2002, 2008a)). Particle abundance data was also used to estimate aerobic (Kalvelage et al., 2015; Thomsen et al., 2019) and anaerobic

respiration (Bianchi et al., 2018; Karthäuser et al., 2021), to inform model development (Bianchi et al., 2018; Stemmann et al., 2004; Jouandet et al., 2014) or calibrate biogeochemical models (Niemeyer, 2020). Changes in the particle distribution were related to physical processes such as transport along continental margins (Stemmann et al., 2008b; Forest et al., 2013; de Madron et al., 1999), deep resuspension (Puig et al., 2013; de Madron et al., 2017) and mesoscale processes (Waite et al., 2016; Fiedler et al., 2016; Stemmann et al., 2008b; Guidi et al., 2012). Profound changes in bacterial activity at Oxygen

minimum zone boundaries (Roullier et al., 2014) were related to enhanced particle abundance. Likewise, the importance of phyto- (Stemmann et al., 2002; Guidi et al., 2009) and zooplankton (Hauss et al., 2016; Christiansen et al., 2018; Stemmann et al., 2004) interactions with particles were assessed and the introduction of particles at depth via zooplankton Diel Vertical Migration reported (Kiko et al., 2017, 2020; Stemmann et al., 2000). In recent years, image analysis of large objects was performed and plankton organisms were discriminated from the detrital and abiotic particles in subsets of the dataset presented

here, allowing the study of large plankton communities (Forest et al., 2012), bio-geography of specific taxa (Christiansen et al., 2018; Biard et al., 2016), zooplankton functional traits (Vilgrain et al., 2021) and particle types (Trudnowska et al., 2021).

### 1.4 The global marine particle size distribution dataset

Here we provide a dataset that was obtained with several inter-calibrated UVP5 units operated by different laboratories and during different cruises and projects around the world (Table 1). This international, collaborative effort resulted in a consistent,

inter-calibrated global marine particle size distribution database that contains 8805 particle abundance and biovolume profiles obtained in all major oceans and several marginal seas since 2008. We provide further details about the UVP5, the inter-calibration and quality control procedures and the dataset structure in the Material and Methods section. Summarizing statistics, maps on data distribution, a description of global particle distribution and recommendations for use and further growth of the dataset are provided in the Results and Discussion sections.



## 2 Material and Methods

### 2.1 UVP5 description

The UVP5 consists of one downward-facing camera in a titanium pressure case and two sets of red LED lights that illuminate a 0.88 to 1.16 L-water volume Picheral et al. (2010). The imaged volume depends on the actual instrument set-up which was determined experimentally for each set-up. During deployment - usually during the downcast of a CTD profile - the UVP5 takes 5-20 pictures of the illuminated volume of water per second. The particles in each image are counted and sized immediately and the data are stored in the instrument. Particle area is measured as the number of pixels ($Sp$) of an imaged object and can be converted to particle cross-sectional area ($Sm$) in $mm^2$ using: $Sm = Aa * Sp^{Exp}$. Here, $Aa$ represents the area of one pixel in $mm^2$. $Exp$ is a dimensionless adjustment factor. $Aa$ and $Exp$ need to be calibrated experimentally. To conduct the initial calibration for the dataset provided here, natural plankton and particle objects from the Bay of Villefranche-sur-Mer, France, were imaged in an aquarium with the UVP5 HD sn203 and using a stereo microscope during experiments conducted in fall 2016. Optimal values for the parameters $Aa$ and $Exp$ were obtained by minimizing $\Delta S$, defined as

$$\Delta S = \sum_i (\log(Aa \cdot S_{i,p_u}^{Exp}) - \log(S_{i,m_\mu}))^2$$

where $S_{i,p_u}$ is the surface area in pixels of object $i$ as seen by the UVP and $S_{i,m_\mu}$ is the surface area in mm$^2$ of the same object $i$ measured under the stereomicroscope. The minimization was performed using the Nelder-Mead simplex algorithm (implemented in the MATLAB function *fminsearch*). For this calibration experiment an $Aa$ of 0.0036 $mm^2$ (interquartile range from -0.0002 to 0.0074 $mm^2$), an $Exp$ of 1.149 (interquartile range 1.016 to 1.282) and an $r^2$ of 0.88 were found. Further details regarding the initial calibration procedure for the UVP5 SD version that was also applied to obtain the UVP5 HD calibration coefficients are given in Picheral et al. (2010).

### 2.2 Instrument inter-calibration

As several UVP5 units were produced, an inter-calibration procedure was developed to allow comparability of data from these units. The inter-calibration procedure is based on a comparison between one or several reference units (in particular sn002 and sn203) and the units to be calibrated. The imaged volume of each unit is determined, before the instruments are deployed at sea simultaneously on the same instrument carrier and the normalized size spectra are calculated. These operations have been performed since 2008 in the Mediterranean Sea off Nice, France. Figure 1a shows an example of raw data from a parallel deployment of a reference unit (sn002) and a unit under calibration (sn200). The $Aa$ and $Exp$ of the sn200 is then adjusted, so that after post-processing the normalized size spectra of both units coincide (Figure 1b). The reference units were regularly inter-calibrated against each other to check for possible drifts and improved data consistency. The development of the HD version of the UVP5 in 2016 required a revision of the UVP5 inter-calibration procedure, as pixel resolution has changed (Picheral et al., 2010). The calibration obtained for the HD unit sn203 in fall 2016 was propagated to several STD model reference units via simultaneous deployment at sea and subsequent calculation of correction factors. Thereafter, the corrections



obtained for the reference units were digitally propagated to all previously used UVP5 by reanalysing the earlier calibration
experiment data. How the uncertainties of the initial calibration of the HD model sn203 propagate to the other UVP5 units and
if these uncertainties can be reduced need to be further investigated.

## 2.3 Data collection, processing, quality control and dataset description

Metadata (position, time) of each profile collected in the presented dataset were checked by the respective data owners.
All instrument settings and calibration coefficients for all cruises and projects were checked and, if necessary, corrected to
match the HD inter-calibration results using automatic routines. Data from all cruises were then reprocessed using Zoopro-
cess (https://sites.google.com/view/piqv/zooprocess) to obtain a coherent and inter-calibrated dataset, based on the HD inter-
calibration conducted in fall 2016. For easier access and preliminary sharing, the data were then uploaded to EcoPart (http://ecopart.obs-
vlfr.fr). To enable the archiving at Pangaea, data were directly downloaded from the EcoPart SQL database using a dedicated
Python script and separated into three-year splits to obtain smaller file sizes and to enable the subsequent addition of further
data.

During processing, the silhouette area of each particle is calculated as described above and then converted to an equivalent
spherical diameter (ESD) according to $ESD = \sqrt{4 * Sm/\pi}$. Biovolume is calculated assuming a spherical particle using
$Biovolume = ESD^3 * \pi/6$. Particles in a certain size class (e.g., ESD: 0.0403 - 0.0508 mm) and within a 5 dbar depth range
are then counted and divided by the total observed volume to yield the particle abundance (#/L) in this size and depth interval.
Likewise, the biovolume of individual particles is added up and divided by the observed volume to yield biovolume in $mm^3/L$.
Size class bins are evenly spaced in a natural logarithmic scale, starting at 0.001 mm and ending at 26 mm, with in total 45 size
bins. Size class bin width is hence increasing with size in a logarithmic fashion. Due to the detection limits of the UVP5, size
class bins smaller 0.0403 mm ESD are empty and not reported, the largest size bin covers the size range from 20.6 to 26 mm
ESD. Particle abundance and biovolume of particles with an ESD > 26 mm is also provided as an additional value. Data in this
form is available on the EcoPart server. Quality-checked data was downloaded from the server on May 26, 2021 and submitted
to Pangaea (https://doi.pangaea.de/10.1594/PANGAEA.924375). Apart from the particle abundance and biovolume in different
size classes, the dataset contains the cruise id, the EcoPart Project identifier (integer), the Profile identifier, the filename of the
150 raw file, the filename of an accompanying ctd profile (if this exists), latitude, longitude (both in decimals), date and time (in
UTC), an EcoPart internal station identifier (integer), depth (indicated via the middle value of the 5 dbar depth bin; in dbar) and
the observed volume per depth bin (in L). The particle size distribution data reported is inclusive of all living and non-living
particles across the size range of detection. The dataset (available at https://doi.pangaea.de/10.1594/PANGAEA.9243751) con-
tains all individual profile data. Also the values of $AA$, $Exp$ and the imaged volume for each data acquisition are archived at
155 Pangaea in the "Metadata collection for a global marine particle size distribution dataset obtained with the Underwater Vision
Profiler 5" (available at: https://download.pangaea.de/reference/106293/attachments/Project_metainfo_for_Pangaea-4.txt), to-
gether with the dataset presented here. To enable visualization within this article, we aggregated particle abundance as in Kiko
et al. (2017) into micrometric- (MiP: 0.14 to 0.53 mm ESD) and macroscopic particles (MaP: 0.53 mm to 16.88 mm ESD), and



calculated the slope $k$ of the differential particle size distribution as a descriptor of the relationship between particle abundance and size (Stemmann and Boss, 2012). This relationship is generally approximated by a two-parameter power-law function: $N = bd^{-k}$, where $b$ and $k$ are constants and $d$ is the mean particle diameter for a given diameter range $(dr)$. The differential particle abundance $N$ can be calculated as the total number of objects per unit volume in the given diameter range $dr$ (e.g. 0.203 mm to 0.256 mm) divided by the diameter range (in this case 0.053 mm) and is given as the number of particles per volume per size. To obtain an estimate of $k$, which is also referred to as the slope of the particle size distribution, one can then conduct a linear regression of $\log(N)$ vs. $\log(d)$ as $\log(N) = \log(b) - k(\log(d))$. The PSD slope $k$ is calculated for the size range 0.203 mm to 2.05 mm, as this is the size range where the slope is mostly linear. These slope $k$ is only considered if the p-value of the regression is $< 0.05$, otherwise the value is set to *nan* (not a number).

Already published datasets (Table 2) use different calibration coefficients which are not consistent with the HD inter-calibration procedure and differences may arise when comparing the different versions. As an example we calculated abundances of two size classes and spectral slopes using the datasets from RV Maria S Merian cruise MSM23 and several RV Meteor cruises. MiP abundances are 4.2 (median; interquartile range 3.8 to 4.7) times larger with the new calibration factors, whereas MaP abundances are 1.5 times larger (median; interquartile range 1.2 to 2.0). Estimates of the slope $k$ of the PSD are 1.09 (median; interquartile range 1.05 to 1.12) times larger. These factors were calculated using the datasets from RV Maria S Merian cruise MSM23 and RV Meteor cruises M92, M96 and M107 for which archived datasets with the relevant data exist.

We do not distinguish UVP5 particle data into distinct categories, such as copepods, aggregates, fecal pellets or other taxonomic or morphologic classes. For UVP5 data, this is possible for objects > 1 mm ESD, as the UVP5 also retrieves "vignettes" - small thumbnail images of respective regions of interest. Homogeneous identification of these vignettes among different cruises and operators is a time-consuming task and was not yet achieved for the entire dataset. Data from a subset of profiles are currently being prepared for publication.

# 3   Results and Discussion

## 3.1   Data distribution

The global distribution of UVP5 profiles contained in the published dataset is shown in Figure 2; in total, it comprises 8805 profiles, collected between 2008-06-19 and 2020-11-23 and between 81.3695 °N and 75.289 °S. The dataset represents a compilation of particle data from numerous small regional-scale research cruises as well as several large-scale hydrographic transects with bathypelagic and cross-basin coverage. All major ocean basins, the Mediterranean and the Baltic Sea were sampled. Most data is available from the Mediterranean Sea, the tropical Atlantic and Pacific, the Gulf of Alaska and the Arctic. Information on the number of profiles obtained per year, month and depth level is shown in Figure 3. The majority of profiles was collected in the upper 1000 m of the water column in June and August. Between 217 and 1146 profiles per year were obtained between 2008-2018. Almost all UVP5 data obtained between 2008 and 2019 are contained in our dataset. We were not able to obtain data from all UVP5 owners and can therefore not provide an exact estimate of how many profiles are currently missing from the dataset. Furthermore, some datasets obtained in 2019 and 2020 still require processing and will be





added in subsequent updates of the dataset. Sampling effort is biased to the Northern hemisphere summer. Of all 8805 profiles, 1676 (19%) are shallower than 200 dbar, 7127 (80%) cover the upper 200 dbar of the water column, 3426 (38%) the upper 1000 dbar and 1018 (11%) go down to at least 3000 dbar. Deep profiles are mostly full depth profiles. The deepest profile

reached 6017.5 dbar depth. Figure 4 shows the maximum depth per 2 degree grid box, whereas figure 5 shows the number of profiles obtained per 2 degree grid box.

## 3.2 Global particle abundance patterns

The global UVP5 particle dataset enables the characterisation of particle abundance and size structure patterns at a global scale, but also enables specific insights into particle dynamics at several regional study sites (e.g., the Gulf of Alaska, the

200 California, Humboldt, Benguela and Mauretania upwelling systems and the Mediterranean Sea). Here, we aim to provide a short description of global particle distribution patterns and reference a few, already published studies. We use the terms micrometric particles (MiP) for particles with 0.14 to 0.53 mm ESD and macroscopic particles (MaP) for particles with 0.53 mm to 16.88 mm diameter) as in Kiko et al. (2017). Thereby, we also follow an approach used for marine aggregates, where those larger than 0.5 mm ESD are defined as marine snow (Suzuki and Kato, 1953; Alldredge and Silver, 1988). Globally, MiP

and MaP concentrations in the upper 200 m are very variable (Figures 6, 9). High MiP and MaP particle abundance in coastal regions, in upwelling or frontal zones (MiP maximum values > 50000 #/L, MaP maximum values > 2000 #/L) are likely due to higher biological production and coastal inputs (Guidi et al., 2008a; Stemmann et al., 2008b; Roullier et al., 2014; Kiko et al., 2017). Particle concentrations are lower in oligotrophic gyres (MiP minimum values: 0.81 #/L MaP minimum values, 0.0 #/L MaP) where productivity and advective input from coastal regions are low (Guidi et al., 2008a, 2009; Stemmann et al.,

2008a; Guidi et al., 2015). Particle abundance generally declines from the surface to depth (compare Figures 5, 6 and 7, as well as 8, 9 and 10). MiP and MaP in the meso- and bathy-pelagic layers also show a pattern consistent with the upper surface pattern probably as a consequence of passive flux of sinking particles (Guidi et al., 2015) and the active supply of particles via diel vertical migrations of zooplankton and nekton to the mesopelagic (Kiko et al., 2017, 2020). The strength of these supply mechanisms is dependent on the biological productivity at the surface, the strength of the active transport processes and the

attenuation processes in the mesopelagic (Guidi et al., 2009). For the following analyses of vertical particle distribution in the open ocean, we only use data from profiles that were conducted down to at least 3000 dbar. For this subset, we find that MiP concentrations range from 0.81 to 53486.0 #/L between 0-200 dbar (mean: 315.67, std: 1269.53), 1.29 to 38580.0 #/L between 200-1000 dbar (mean: 54.39, std: 228.86) and 0.7 to 3184.0 #/L between 1000-3000 dbar (mean: 15.47, std: 23.76). MaP concentrations ranging from 0.0 to 2130.1 #/L between 0-200 dbar (mean: 6.17, std: 47.52), 0.0 to 2560.0 #/L between

200-1000 dbar (mean: 0.89, std: 5.17) and 0.0 to 77.77 #/L between 1000-3000 dbar (mean: 0.16, std: 0.57). The decline of particle abundance with depth has been interpreted as a consequence of microbial and metazoan flux attenuation (Stemmann et al., 2004; Guidi et al., 2009). The variability in MiP and MaP abundance range also decreases from the epi- to the bathy-pelagic, suggesting a feedback mechanism where high particle abundance results in strong flux attenuation by metazoans, thereby removing peaks in particle abundance and flux (Guidi et al., 2009).





### 3.3 Slope of the particle size distribution

The size distribution of particles is a basic property of marine systems, affecting trophic interactions, the vertical transmission of solar energy and the downward transport of organic matter Stemmann and Boss (2012). Despite its fundamental importance, size distribution is difficult to measure because particles occur over a large range of size and composition, from sub-micrometer compact particles to large, cm-sized loose aggregates (Jackson et al., 1995; Stemmann et al., 2008a; Lombard et al., 2019). We here use the differential particle size distribution as e.g. described by Stemmann and Boss (2012). A slope $k$ of 4 of the differential particle size distribution suggests an equal amount of mass in logarithmic increasing size intervals. By combining instruments over a $\mu$m to cm size range it was shown that the value of the slope varies greatly around the typical value of 4 Jackson et al. (1995); Stemmann et al. (2008a). Our study also shows that the slope $k$ varies greatly in the epi-, meso- and bathypelagic (Figures 12, 13 and 14). If we constrain the dataset to profiles that go deeper than 3000 dbar, the global mean value of the slope $k$ in the top 200 dbar of the water column is found to be -3.57 +/- 0.56 std (minimum -6.58, maximum -1.8), with significant variations from -4 which are likely due to local ecosystem processes and other impacts. The average slope $k$ and the variability remain similar at greater depth ( -3.59 +/- 0.67 std, minimum -8.25, maximum -1.37 at 200-1000 dbar depth, -3.52 +/- 0.6 std, minimum -7.34, maximum -1.33 at 1000 to 3000 dbar depth). Throughout all depth ranges, steepest slopes are observed in oligotrophic basins such as the Eastern Mediterranean Sea and the center of the South Pacific gyre, while flatter spectra are observed in more productive regions such as the Western Mediterranean Sea and at high latitudes. These observations confirm earlier work using more restricted datasets (Guidi et al., 2009; Stemmann et al., 2008c; Guidi et al., 2008a). Earlier work based on a sub-sample of the dataset has also shown that the slope of the size spectrum is correlated with the phytoplankton community composition (Guidi et al., 2009; Stemmann et al., 2002) and can show diel variability related to zooplankton migration (Stemmann et al., 2000). Deeper in the water column, the spatial pattern of the slope $k$ mostly reflects the upper ocean variability. Interestingly, bathypelagic values of $k$ in the Antarctic are relatively flat, compared to temperate and tropical regions, which suggests that, in the Antarctic deep sea the relative role of larger, aggregated particles is more important than in the temperate and tropical regions. Such trend is not observed in data from Arctic regions.

### 3.4 Potential uses of the data

A further, detailed analysis of the provided dataset is beyond the scope of this article. Observation of a particle at a certain depth always generates the question how it was formed or arrived at the given location. Many attempts have been carried out to relate the UVP particle size spectrum with flux measured in sediment traps or by Thorium (Guidi et al., 2008b; Forest et al., 2013; Guidi et al., 2015), sinking speed (Stemmann et al., 2002) and POC (Stemmann et al., 2008a) but deriving biogeochemical properties from particle size is certainly an area for future progress. In these regards, our dataset should enable further regional and global analyses of particle dynamics (see e.g. Bisson et al. (2021)) and - in combination with flux estimates from sediment traps and/or Thorium isotope measurements, but also environmental data from satellites and other sources - enable us to better constrain the particle flux component of the biological pump (see e.g., Clements et al. (2021b, a)). However, we would like to stress that although the particles in our dataset are not per se sinking (e.g., also living zooplankton are treated as particles),





particle abundance and size alone are still important information. Therefore, the data is also especially useful to constrain models that explicitly generate a particle size spectrum (Bianchi et al., 2018; Niemeyer, 2020; Weber and Bianchi, 2020;
Stemmann et al., 2004; Jouandet et al., 2014). On the other hand particle data can also be used to estimate remineralization rates (Kalvelage et al., 2015; Bianchi et al., 2018; Thomsen et al., 2019; Karthäuser et al., 2021) or study trace element scavenging.

## 3.5 Recommendations for further instrument usage and growth of the dataset

This work presents the first attempt to establish a calibrated global data set of UVP measurements. Our analysis led us to
265 develop a set of recommendations for future expansion of the global UVP data set. First of all, we recommend that full depth profiles are always taken at locations shallower than 1000 m depth and that otherwise at least the full mesopelagic down to 1000 m depth be sampled when using the UVP5. This is motivated by the fact that particle processes (indicated via a large range of e.g., MiP and MaP abundance) at these depths are very dynamic and require high resolution sampling. Below 1000 m depth, particle spatial patterns are less variable. Nevertheless, if sampling during a research cruise is conducted at water depths
> 1000 m, full depth profiles or profiles down to the maximum depth rating of the used instruments (typically 6000m) should be done as often as possible. The deep sea is not well characterized with respect to abundance and size of particles and these comparatively small demands on shiptime will generate an important added value, as this will e.g. enable us to further assess carbon sequestration in the deep sea. Regions that are not well sampled until now are the Indian Ocean, Antarctic waters and the Western Pacific. Furthermore, winter data from both hemispheres is mostly lacking as well. In general, the UVP should be
used during repeat hydrography programs as the operational goals of these programs to cover a representative fraction of the ocean (global and full depth coverage) align with our goals to create a global particle size distribution dataset. SCOR working group 154 "Integration of Plankton-Observing Sensor Systems to Existing Global Sampling Programs (P-OBS)" recommended the use of the UVP during the GO-ship program and similar sea-going expeditions (Boss et al., 2020). Data from the smaller and more versatile UVP6 (Picheral et al., 2022) that can also be deployed on gliders, floats, moorings and other vectors should
also be integrated in future datasets and will enable the study of particle dynamics at spatial and temporal scales that are not accessible with the UVP5. Ancillary data that is useful for the interpretation of UVP data are temperature, salinity, oxygen and nutrient measurements, measurements of current dynamics, but also any measurements of particle dynamics and characteristics (e.g., Thorium-isotope measurements, lipid-content, elemental composition, particle sinking speed, sedimentation flux) and data on bacterio- phyto- and zooplankton composition. The latter are especially needed to understand the ecological processes
behind the observed size spectra of particles and their subsequent export. The evaluation of the relative proportions of living and non-living particles is particularly important at the large size range (few hundreds of micrometer) because large, possibly sinking particles may be confused with zooplankton and lead to overestimation of particle stock and flux (Kiko et al., 2020). In the future, better automatic image classification algorithms may help to discriminate between non-living particles and plankton organisms and even provide information on other properties than their size (Stemmann and Boss, 2012; Trudnowska et al.,
2021). We strongly recommend that regular inter-calibration experiments of all instruments against one or several standard units take place to maintain the data quality of all UVP units at an inter-operable level.





## 4   Conclusions

Here we provide the first global particle size spectra dataset containing 8805 profiles that were obtained with the UVP5 between
2008 and 2020. All UVP5s used were inter-calibrated with a standard procedure, calibration coefficients and metadata were
checked and all profile data were reprocessed. This dataset therefore is internally consistent and supersedes earlier versions of
cruise-specific UVP5 particle size spectrum data. The analysis of this global dataset shows that particle abundances are high
in regions of high primary productivity and in coastal areas. Further analysis of the dataset should enable insights on different
aspects of particle dynamics such as the effects of mesoscale features and Oxygen Minimum Zones, the fate of particulate
matter in the deep sea and many other important aspects of the oceans biogeochemistry.

## 5   Data availability

The global UVP5 particle dataset Kiko et al. (2021) is publicly available at https://doi.pangaea.de/10.1594/PANGAEA.924375.
The dataset was downloaded from the https://ecopart.obs-vlfr.fr/ server on the 15.February 2022.

*Author contributions.*

RK, MP and LS formulated the goals for data aggregation, quality control and the publication of a global UVP5 dataset.
RK and MP led the quality control endeavors, supported by all co-authors. RK and LS conceived and drafted the article. All
authors participated writing the article.

*Competing interests.*

The authors declare no competing interest.

*Acknowledgements.*  RK acknowledges support via a "Make Our Planet Great Again" grant of the French National Research Agency within
the "Programme d'Investissements d'Avenir"; reference "ANR-19-MPGA-0012" and via EU H2020 grant (agreement 817578 TRIATLAS
project). RK and HH furthermore acknowledge support by the DFG-funded collaborative research center 754 "Climate-biogeochemistry
interactions in the tropical Ocean" (Work Package B8) and the "CUSCO—Coastal Upwelling System in a Changing Ocean" project (Grant
no. 03F0813A; Work package 5) funded by the Federal Ministry of Education and Research (Germany). The McDonnell Laboratory at the
University of Alaska Fairbanks acknowledges support from the U.S. National Science Foundation (Award numbers 1654663 and 1656070),
the U.S. National Aeronautics and Space Administration (Award number 80NSSC17K0692), and the M.J. Murdock Charitable Trust. The
collection of data by the University of Maine were funded by NASA grants NNX15AE67G (NAAMES) and 80NSSC17K0568 (EXPORTS).
We thank the support of CNRS-INSU through the MISTRALS-MERMEX program. LS was supported by the Chair Vision between CNRS
and Sorbonne University.



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





**Tables**

**Table 1**

Table 1: Geospatial information for UVP projects.

| UVP project name | Ecopart ID | Profiles | Time period | Latitude range | Longitude range | UVP manager(s) |
|---|---|---|---|---|---|---|
| uvp5_sn000_boum2008 | 2 | 184 | 2008-06-19 to 2008-07-18 | 43.21 to 33.47 | 32.77 to 4.93 | L Stemmann, M Picheral |
| uvp5_sn000_ccelter_2012 | 3 | 62 | 2012-07-28 to 2012-08-21 | 34.6 to 33.1 | -118.31 to -123.69 | L Stemmann, M Picheral |
| uvp5_sn000_lohafex2009 | 4 | 57 | 2009-01-16 to 2009-03-06 | -47.58 to -50.01 | -13.64 to -35.26 | L Stemmann, M Picheral |
| uvp5_sn000_lter2008 | 5 | 75 | 2008-09-30 to 2008-10-28 | 34.2 to 32.15 | -117.96 to -124.0 | L Stemmann, M Picheral |
| uvp5_sn000_malina2009 | 16 | 154 | 2009-07-18 to 2009-08-22 | 72.06 to 69.47 | -126.48 to -140.81 | L Stemmann, M Picheral |
| uvp5_sn000_msm049 | 19 | 22 | 2015-11-30 to 2015-12-19 | 20.32 to 12.0 | -20.5 to -24.28 | R Kiko, H Hauss |
| uvp5_sn000_operex2008 | 20 | 92 | 2008-07-31 to 2008-08-12 | 25.75 to 22.24 | -156.25 to -160.67 | L Guidi, L Stemmann |
| uvp5_sn000_tara2009 | 6 | 46 | 2009-10-11 to 2009-12-15 | 43.36 to 33.37 | 35.33 to 7.89 | L Stemmann, M Picheral |
| uvp5_sn000_tara2010 | 7 | 196 | 2010-01-09 to 2010-12-17 | 27.16 to -55.1 | 73.91 to -65.91 | L Stemmann, M Picheral |
| uvp5_sn000_tara2011 | 99 | 264 | 2011-01-03 to 2011-12-21 | 35.42 to -64.36 | -53.01 to -159.06 | L Stemmann, M Picheral |
| uvp5_sn000_tara2012 | 9 | 32 | 2012-01-28 to 2012-02-18 | 39.24 to 32.92 | -66.54 to -75.07 | L Stemmann, M Picheral |
| uvp5_sn001_2012_moose_ge | 21 | 87 | 2012-07-24 to 2012-08-08 | 43.9 to 40.0 | 9.64 to 3.44 | L Stemmann, M Picheral |
| uvp5_sn001_2012_msm22 | 22 | 113 | 2012-10-24 to 2012-11-22 | 18.5 to -5.01 | -19.68 to -26.99 | R Kiko, H Hauss |
| uvp5_sn001_2012_msm23 | 23 | 64 | 2012-11-26 to 2012-12-16 | 17.6 to -18.19 | 1.0 to -24.3 | R Kiko, H Hauss |
| uvp5_sn001_2013_m92 | 24 | 30 | 2013-01-19 to 2013-01-30 | -11.0 to -12.61 | -77.17 to -78.63 | R Kiko, H Hauss |
| uvp5_sn001_2013_m93 | 25 | 148 | 2013-02-08 to 2013-03-04 | -12.16 to -13.97 | -76.42 to -78.42 | R Kiko, H Hauss |
| uvp5_sn001_2013_m96 | 26 | 77 | 2013-05-02 to 2013-05-22 | 17.7 to 11.33 | -20.08 to -60.3 | R Kiko, H Hauss |
| uvp5_sn001_2013_m97 | 27 | 180 | 2013-05-26 to 2013-06-23 | 17.57 to 8.0 | -17.75 to -24.28 | R Kiko, H Hauss |
| uvp5_sn001_2013_m98 | 28 | 52 | 2013-07-02 to 2013-07-23 | -5.12 to -11.5 | 13.5 to -35.89 | R Kiko, H Hauss |
| uvp5_sn001_2014_msm40 | 29 | 5 | 2014-08-17 to 2014-08-19 | 59.54 to 59.19 | -39.74 to -43.54 | R Kiko, H Hauss |
| uvp5_sn002_iado_2014 | 251 | 26 | 2014-09-20 to 2014-09-23 | 43.69 to 43.37 | 7.89 to 7.14 | J-O Irisson |
| uvp5_sn002_iado_2015 | 252 | 36 | 2015-09-16 to 2015-09-20 | 43.65 to 43.42 | 7.8 to 7.13 | J-O Irisson |
| uvp5_sn002_iado_2016 | 30 | 16 | 2016-09-18 to 2016-09-21 | 43.67 to 43.39 | 7.6 to 7.31 | J-O Irisson |
| uvp5_sn002_iado_2018 | 121 | 10 | 2018-09-22 to 2018-09-23 | 43.59 to 43.31 | 7.68 to 7.39 | J-O Irisson |
| uvp5_sn002_moose_dyf_2013 | 10 | 4 | 2013-09-14 to 2013-10-24 | 43.42 to 43.42 | 7.9 to 7.9 | L Stemmann, M Picheral |
| uvp5_sn002_moose_dyf_2014 | 11 | 9 | 2014-03-11 to 2014-12-10 | 43.68 to 43.36 | 7.9 to 7.31 | L Stemmann, M Picheral |
| uvp5_sn002_moose_dyf_2015 | 12 | 9 | 2015-02-08 to 2015-12-10 | 43.44 to 43.42 | 7.87 to 7.82 | L Stemmann, M Picheral |
| uvp5_sn002_moose_dyf_2016 | 13 | 10 | 2016-02-05 to 2016-12-10 | 43.43 to 43.41 | 7.87 to 7.86 | L Guidi |
| uvp5_sn002_moose_dyf_2017 | 14 | 8 | 2017-02-07 to 2017-11-08 | 43.43 to 43.41 | 7.88 to 7.86 | L Coppola |
| uvp5_sn002_moose_dyf_2018 | 166 | 4 | 2018-01-23 to 2018-08-27 | 43.42 to 43.41 | 7.87 to 7.85 | L Guidi |
| uvp5_sn002_moose_ge_2013 | 15 | 6 | 2013-06-11 to 2013-06-15 | 43.75 to 43.41 | 9.36 to 7.52 | L Stemmann, M Picheral |

510

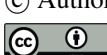



| Station | | | Date range | Latitude | Longitude | PI |
|---|---|---|---|---|---|---|
| uvp5_sn002_moose_ge_2014 | 17 | 84 | 2014-07-04 to 2014-07-20 | 43.93 to 40.0 | 9.72 to 3.5 | L Stemmann, M Picheral |
| uvp5_sn002_moose_ge_2015_filtered | 31 | 72 | 2015-07-10 to 2015-07-27 | 43.88 to 40.0 | 9.63 to 3.54 | L Stemmann, M Picheral |
| uvp5_sn002_moose_ge_2016_filtered | 18 | 84 | 2016-05-19 to 2016-06-09 | 43.63 to 40.0 | 8.92 to 3.54 | L Guidi |
| uvp5_sn002_moose_ge_2017_filtered | 100 | 116 | 2017-08-31 to 2017-09-23 | 43.88 to 39.99 | 9.63 to 3.5 | L Coppola |
| uvp5_sn002_moose_ge_2019 | 149 | 88 | 2019-06-08 to 2019-07-01 | 43.88 to 40.0 | 9.63 to 3.5 | L Coppola |
| uvp5_sn002_somba_ge_2014 | 32 | 65 | 2014-08-17 to 2014-09-08 | 39.72 to 36.55 | 9.48 to -0.7 | L Stemmann, M Picheral |
| uvp5_sn002zd_cascade2011 | 33 | 82 | 2011-03-02 to 2011-03-21 | 43.4 to 41.13 | 6.13 to 3.36 | X. Durrieu de Madron |
| uvp5_sn002zd_ccelter_2011 | 34 | 58 | 2011-06-27 to 2011-07-16 | 34.11 to 32.58 | -120.85 to -121.78 | L Stemmann, M Picheral |
| uvp5_sn002zd_gatekeeper2010 | 35 | 21 | 2010-07-11 to 2010-07-14 | 36.8 to 36.7 | -121.97 to -122.58 | L Stemmann, M Picheral |
| uvp5_sn002zd_keops2 | 36 | 106 | 2011-10-14 to 2011-11-20 | -45.0 to -50.65 | 75.0 to 52.1 | L Stemmann, M Picheral |
| uvp5_sn002zd_keops2 | 36 | 106 | 2011-10-14 to 2011-11-20 | -45.0 to -50.65 | 75.0 to 52.1 | L Stemmann, M Picheral |
| uvp5_sn002zd_omer | 37 | 10 | 2012-04-08 to 2012-04-09 | 43.61 to 43.58 | 7.58 to 7.49 | L Stemmann, M Picheral |
| uvp5_sn002zd_omer_2 | 38 | 7 | 2012-05-20 to 2012-05-22 | 43.69 to 43.69 | 7.32 to 7.31 | L Stemmann, M Picheral |
| uvp5_sn003_2015_kaxis | 40 | 3 | 2016-01-21 to 2016-01-23 | -61.97 to -62.7 | 95.37 to 91.53 | L Stemmann, M Picheral |
| uvp5_sn003_cassiopee_2015 | 41 | 82 | 2015-07-20 to 2015-08-15 | 2.0 to -19.98 | 168.01 to 148.05 | L Stemmann, M Picheral |
| uvp5_sn003_ccelter_2014 | 42 | 62 | 2014-08-07 to 2014-09-02 | 34.87 to 32.28 | -118.28 to -123.9 | L Stemmann, M Picheral |
| uvp5_sn003_ccelter_2014 | 42 | 62 | 2014-08-07 to 2014-09-02 | 34.87 to 32.28 | -118.28 to -123.9 | L Stemmann, M Picheral |
| uvp5_sn003_ccelter_2016 | 43 | 60 | 2016-04-20 to 2016-05-11 | 35.09 to 32.7 | -117.36 to -123.21 | L Stemmann, M Picheral |
| uvp5_sn003_csiro_iioe | 144 | 52 | 2019-05-15 to 2019-06-09 | -11.45 to -39.49 | 113.42 to 109.88 | T Biard |
| uvp5_sn003_dewex_spring_2013 | 44 | 83 | 2013-04-05 to 2013-04-19 | 43.63 to 40.08 | 8.64 to 3.51 | D Antoine |
| uvp5_sn003_iado_2017 | 90 | 24 | 2017-09-22 to 2017-09-24 | 43.67 to 43.35 | 7.65 to 7.31 | L Stemmann, M Picheral |
| uvp5_sn003_jerico_2017 | 1 | 27 | 2017-07-10 to 2017-07-16 | 59.85 to 54.97 | 24.84 to 10.5 | J-O Irisson |
| uvp5_sn003_mobydick_2018 | 124 | 61 | 2018-02-21 to 2018-03-19 | -29.04 to -52.6 | 74.9 to 59.06 | L Stemmann, M Picheral |
| uvp5_sn003_outpace_2015 | 46 | 205 | 2015-02-21 to 2015-03-31 | -17.9 to -22.0 | 178.64 to -178.51 | L Guidi |
| uvp5_sn003_sargasso_a | 47 | 52 | 2014-03-16 to 2014-04-05 | 31.5 to 24.67 | -62.48 to -68.56 | L Guidi |
| uvp5_sn003_sargasso_b | 48 | 32 | 2014-04-09 to 2014-04-22 | 35.18 to 25.67 | -31.63 to -59.52 | F Lombard |
| uvp5_sn003_tara2013 | 49 | 155 | 2013-05-26 to 2013-10-27 | 79.67 to 54.41 | 174.99 to -168.66 | F Lombard |
| uvp5_sn003zp_pelgas2012 | 50 | 34 | 2012-05-26 to 2012-06-03 | 46.11 to 44.86 | -1.27 to -2.6 | L Stemmann, M Picheral |
| uvp5_sn003zp_tara2012 | 51 | 77 | 2011-12-30 to 2012-03-26 | 44.36 to 9.84 | -10.07 to -88.49 | L Stemmann, M Picheral |
| uvp5_sn005_batman | 52 | 6 | 2016-03-11 to 2016-03-15 | 42.8 to 42.8 | 6.08 to 6.08 | F Carlotti, Leo Berline |
| uvp5_sn005_dewex_2013_winter | 53 | 53 | 2013-02-03 to 2013-02-18 | 42.88 to 40.08 | 8.59 to 3.45 | L Stemmann, M Picheral |
| uvp5_sn005_dy032_2015_filtered | 54 | 15 | 2015-06-24 to 2015-07-03 | 49.08 to 48.68 | -16.26 to -17.06 | F Carlotti, Leo Berline |
| uvp5_sn005_moose_ge_2013 | 55 | 39 | 2013-06-29 to 2013-07-07 | 43.05 to 39.96 | 8.0 to 3.39 | L Stemmann, M Picheral |
| UVPsn008_2018_leg02c | 152 | 42 | 2018-07-25 to 2018-08-13 | 71.41 to 59.22 | -48.46 to -70.18 | M Babin, M Picheral |
| uvp5_sn008_an1304 | 56 | 101 | 2013-07-29 to 2013-09-15 | 81.28 to 53.8 | -55.43 to -116.96 | M Babin, M Picheral |
| uvp5_sn008_an1405 | 57 | 64 | 2014-07-27 to 2014-08-13 | 81.37 to 68.68 | -57.88 to -108.51 | M Babin, M Picheral |
| uvp5_sn008_an1406 | 58 | 82 | 2014-08-17 to 2014-09-23 | 75.21 to 69.37 | -123.03 to -169.83 | M Babin, M Picheral |
| uvp5_sn008_an1407 | 59 | 11 | 2014-09-30 to 2014-10-08 | 71.12 to 53.8 | -55.44 to -72.26 | M Babin, M Picheral |





| Name | | | | | | |
|---|---|---|---|---|---|---|
| uvp5_sn008_green_2015_icecamp | 60 | 32 | 2015-04-18 to 2015-06-21 | 67.48 to 67.48 | -63.79 to -63.79 | M Babin, M Picheral |
| uvp5_sn008_green_2016_icecamp | 61 | 29 | 2016-03-02 to 2016-07-04 | 67.48 to 67.48 | -63.79 to -63.79 | M Babin, M Picheral |
| uvp5_sn008_ips_amundsen_2018 | 105 | 7 | 2018-07-16 to 2018-07-22 | 69.29 to 67.24 | -60.39 to -64.64 | M Babin, M Picheral |
| uvp5_sn008_subice_2014 | 62 | 228 | 2014-05-15 to 2014-06-20 | 73.27 to 63.95 | -162.0 to -168.95 | L Stemmann, M Picheral |
| uvp5_sn008_uvp_azomp | 161 | 35 | 2019-06-01 to 2019-06-17 | 60.57 to 44.27 | -48.23 to -63.32 | A.M.P. McDonnell |
| uvp5_sn009_pomz | 132 | 29 | 2016-12-27 to 2017-01-13 | 21.36 to 14.0 | -104.63 to -107.83 | A.M.P. McDonnell |
| uvp5_sn009_en_534_mcdonnell | 257 | 10 | 2013-10-24 to 2013-10-27 | 39.81 to 38.38 | -71.01 to -72.91 | A.M.P. McDonnell, J.S. Turner |
| uvp5_sn009_2015_goa | 63 | 70 | 2015-07-17 to 2015-07-30 | 60.3 to 54.64 | -132.86 to -149.47 | A.M.P. McDonnell |
| uvp5_sn009_2015_p16n | 64 | 171 | 2015-04-11 to 2015-06-18 | 56.29 to -16.96 | -149.86 to -153.23 | A.M.P. McDonnell |
| uvp5_sn009_2015_p16n_goa | 65 | 15 | 2015-06-19 to 2015-06-23 | 56.79 to 54.35 | -135.95 to -149.14 | A.M.P. McDonnell, J.S. Turner |
| uvp5_sn009_2016_goa_fall | 146 | 37 | 2016-09-16 to 2016-09-21 | 61.08 to 57.8 | -146.75 to -149.48 | A.M.P. McDonnell |
| uvp5_sn009_2016_goa_spring | 141 | 33 | 2016-04-30 to 2016-05-27 | 60.99 to 57.79 | -147.08 to -149.49 | A.M.P. McDonnell |
| uvp5_sn009_2017_asgard | 112 | 71 | 2017-06-09 to 2017-06-27 | 69.04 to 63.3 | -164.43 to -172.59 | A.M.P. McDonnell |
| uvp5_sn009_2017_sewardline_fall | 142 | 49 | 2017-09-16 to 2017-09-22 | 60.99 to 57.79 | -146.98 to -149.49 | A.M.P. McDonnell |
| uvp5_sn009_2018_asgard_filtered | 234 | 69 | 2018-06-06 to 2018-06-24 | 69.45 to 61.29 | -164.43 to -171.51 | A.M.P. McDonnell |
| uvp5_sn009_2018_nga_fall_filtered | 131 | 60 | 2018-09-12 to 2018-09-21 | 60.25 to 57.21 | -145.5 to -151.39 | A.M.P. McDonnell |
| uvp5_sn009_2019_nga_lter_spring_filtered | 139 | 54 | 2019-04-30 to 2019-05-08 | 60.83 to 56.97 | -147.39 to -151.58 | A.M.P. McDonnell |
| uvp5_sn009_2019_nga_lter_summer_filtered | 151 | 57 | 2019-06-29 to 2019-07-17 | 60.53 to 56.66 | -144.59 to -151.59 | A.M.P. McDonnell |
| uvp5_sn009_2019_nga_lter_summer_filtered | 151 | 57 | 2019-06-29 to 2019-07-17 | 60.53 to 56.66 | -144.59 to -151.59 | A.M.P. McDonnell |
| uvp5_sn009_chukchi borderlands_2016 | 147 | 21 | 2016-07-08 to 2016-08-02 | 78.35 to 71.6 | -158.48 to -164.06 | A.M.P. McDonnell |
| uvp5_sn009_2018_nga_spring_filtered | 104 | 70 | 2018-04-19 to 2018-05-04 | 61.25 to 57.79 | -143.89 to -149.47 | A.M.P. McDonnell |
| uvp5_sn009_sewardline_f2014 | 66 | 10 | 2014-09-13 to 2014-09-16 | 59.84 to 58.24 | -147.93 to -149.49 | A.M.P. McDonnell |
| uvp5_sn009_tb14 | 148 | 24 | 2014-08-20 to 2014-08-28 | 70.62 to 69.72 | -140.3 to -145.11 | A.M.P. McDonnell |
| uvp5_sn009_txs14 | 67 | 9 | 2014-05-03 to 2014-05-05 | 59.84 to 58.68 | -148.35 to -149.48 | A.M.P. McDonnell |
| uvp5_sn010_2014_eddy | 109 | 6 | 2014-02-14 to 2014-03-07 | 19.51 to 16.75 | -24.3 to -25.12 | R Kiko, H Hauss |
| uvp5_sn010_2014_m105 | 68 | 138 | 2014-03-18 to 2014-04-14 | 19.23 to 7.0 | -17.5 to -26.0 | R Kiko, H Hauss |
| uvp5_sn010_2014_m106 | 69 | 115 | 2014-04-19 to 2014-05-24 | 17.6 to -11.5 | -21.21 to -35.89 | R Kiko, H Hauss |
| uvp5_sn010_2014_m107 | 70 | 73 | 2014-06-05 to 2014-06-29 | 19.9 to 11.45 | -16.32 to -23.0 | R Kiko, H Hauss |
| uvp5_sn010_2014_m108 | 71 | 12 | 2014-07-09 to 2014-07-20 | 49.0 to 39.52 | -15.96 to -16.52 | R Kiko |
| uvp5_sn010_2014_ps88b | 72 | 39 | 2014-11-04 to 2014-11-15 | 21.21 to -1.0 | -21.12 to -24.29 | R Kiko, H Hauss |
| uvp5_sn010_2015_m116 | 73 | 82 | 2015-05-02 to 2015-06-02 | 17.58 to 5.0 | -18.0 to -57.67 | R Kiko, H Hauss |
| uvp5_sn010_2015_m119 | 74 | 49 | 2015-09-08 to 2015-09-26 | 17.61 to -5.0 | -21.21 to -24.33 | R Kiko, H Hauss |
| uvp5_sn010_2015_m120 | 75 | 8 | 2015-10-31 to 2015-11-02 | -6.21 to -10.59 | 13.43 to 11.38 | R Kiko, H Hauss |
| uvp5_sn010_2015_m121 | 76 | 88 | 2015-11-22 to 2015-12-24 | -3.0 to -29.58 | 15.56 to -0.01 | R Kiko, H Hauss |
| uvp5_sn010_2016_love | 226 | 43 | 2016-03-30 to 2016-04-07 | 68.27 to 67.78 | 14.7 to 14.04 | H Hauss, R Kiko |
| uvp5_sn010_2016_m130 | 77 | 112 | 2016-08-29 to 2016-10-01 | 17.7 to -11.5 | -19.0 to -35.89 | R Kiko, H Hauss |
| uvp5_sn010_2016_m131 | 223 | 89 | 2016-10-08 to 2016-11-09 | -6.21 to -23.0 | 14.37 to -32.0 | R Kiko, H Hauss |
| uvp5_sn010_2017_fluxes1 | 110 | 72 | 2017-07-14 to 2017-08-08 | 23.0 to 17.5 | -17.64 to -26.0 | R Kiko |





| Name | | | Date range | | | Author |
|---|---|---|---|---|---|---|
| uvp5_sn010_2017_fluxes2 | 111 | 53 | 2017-11-02 to 2017-11-20 | 27.67 to 20.39 | -15.82 to -20.65 | R Kiko |
| uvp5_sn010_2017_m135 | 95 | 141 | 2017-03-02 to 2017-04-07 | -10.67 to -31.03 | -70.3 to -86.0 | R Kiko, H Hauss |
| uvp5_sn010_2017_m136 | 96 | 98 | 2017-04-12 to 2017-05-02 | -12.19 to -15.51 | -76.47 to -78.5 | R Kiko, H Hauss |
| uvp5_sn010_2017_m137 | 97 | 85 | 2017-05-06 to 2017-05-27 | -12.1 to -12.98 | -77.06 to -78.19 | R Kiko, H Hauss |
| uvp5_sn010_2017_m138 | 98 | 42 | 2017-06-03 to 2017-06-29 | 1.5 to -16.25 | -75.43 to -85.84 | R Kiko, H Hauss |
| uvp5_sn010_2018_m145 | 172 | 89 | 2018-02-13 to 2018-03-12 | 17.61 to -11.5 | -21.23 to -35.89 | R Kiko, H Hauss |
| uvp5_sn010_2018_m147 | 171 | 4 | 2018-05-01 to 2018-05-04 | 3.95 to 1.91 | -46.44 to -48.26 | R Kiko, H Hauss |
| uvp5_sn010_2018_m148 | 173 | 92 | 2018-05-30 to 2018-06-28 | -6.21 to -22.67 | 14.21 to -35.88 | R Kiko, H Hauss |
| uvp5_sn011_2016_syTUMSAT1 | 246 | 2 | 2016-09-26 to 2016-09-27 | 35.06 to 35.06 | 138.78 to 138.68 | Y. Tanaka |
| uvp5_sn011_2017_syTUMSAT2 | 247 | 10 | 2017-05-22 to 2017-05-24 | 35.1 to 33.4 | 139.87 to 139.41 | Y. Tanaka |
| uvp5_sn200_ilhas_2017_filtered | 240 | 38 | 2017-02-02 to 2017-02-13 | -20.06 to -21.14 | -28.3 to -40.25 | R. Lopes |
| uvp5_sn200_moose_ge_2018_filtered | 168 | 32 | 2018-05-27 to 2018-06-05 | 43.0 to 40.0 | 7.98 to 3.82 | L Coppola |
| uvp5_sn200_perle_02_2019_filtered | 235 | 31 | 2019-02-27 to 2019-03-04 | 35.95 to 34.04 | 25.3 to 22.96 | X. D. de Madron |
| uvp5_sn200_perle_02_2019_filtered | 235 | 31 | 2019-02-27 to 2019-03-04 | 35.95 to 34.04 | 25.3 to 22.96 | X. D. de Madron |
| uvp5_sn201_2015_naames_01 | 80 | 26 | 2015-11-14 to 2015-11-25 | 54.11 to 40.51 | -37.51 to -40.48 | L. Karp-Boss, E. Boss |
| uvp5_sn201_2016_naames_02 | 81 | 42 | 2016-05-17 to 2016-05-29 | 56.34 to 44.05 | -38.21 to -46.15 | L. Karp-Boss, E. Boss |
| uvp5_sn201_2017_naames_03 | 92 | 40 | 2017-09-04 to 2017-09-17 | 53.4 to 42.38 | -39.13 to -48.95 | L. Karp-Boss, E. Boss |
| uvp5_sn201_2018_naames_04_filtered | 236 | 12 | 2018-03-27 to 2018-04-01 | 44.48 to 39.28 | -38.28 to -43.53 | L. Karp-Boss, E. Boss |
| uvp5_sn201_ccelter_2017 | 83 | 90 | 2017-06-01 to 2017-07-01 | 35.58 to 33.02 | -118.11 to -123.18 | T Biard |
| uvp5_sn201_ccelter_2019_filtered | 154 | 77 | 2019-08-06 to 2019-09-05 | 36.45 to 32.86 | -117.66 to -125.07 | T Biard |
| uvp5_sn201_exports01_filtered | 228 | 84 | 2018-08-14 to 2018-09-09 | 50.6 to 49.93 | -144.35 to -145.22 | L. Karp-Boss, E. Boss |
| uvp5_sn202_msm060_filtered | 231 | 127 | 2017-01-04 to 2017-01-31 | -34.04 to -34.83 | 18.15 to -51.83 | A. Rogge |
| uvp5_sn202_msm074_filtered | 232 | 114 | 2018-05-25 to 2018-06-19 | 60.4 to 47.55 | -36.1 to -54.0 | A. Rogge |
| uvp5_sn202_ps99_20_06_filtered | 237 | 8 | 2016-06-20 to 2016-06-20 | 74.93 to 74.7 | 18.15 to 17.36 | A. Rogge |
| uvp5_sn202_ps99_21_06_3_filtered | 85 | 27 | 2016-06-22 to 2016-07-12 | 79.59 to 77.59 | 11.09 to -5.41 | A. Rogge |
| uvp5_sn203_greenedge_2016 | 86 | 86 | 2016-06-05 to 2016-06-22 | 69.03 to 50.34 | -52.84 to -63.2 | L. Stemmann, M. Picheral |
| uvp5_sn203_greenedge_2016_1b | 87 | 110 | 2016-06-24 to 2016-07-10 | 70.51 to 68.02 | -56.9 to -63.28 | L. Stemmann, M. Picheral |
| uvp5_sn205_coastdark_2019 | 153 | 38 | 2019-07-26 to 2019-08-11 | 79.04 to 76.64 | 16.87 to 7.76 | E. Trudnowska |
| uvp5_sn205_perle_02_2019_filtered | 134 | 81 | 2019-03-04 to 2019-03-16 | 35.88 to 33.54 | 28.81 to 24.38 | X. D. de Madron |
| uvp5_sn205_perle_03_2020_filtered | 238 | 21 | 2020-03-13 to 2020-03-16 | 42.94 to 39.19 | 14.26 to 9.59 | X. D. de Madron |
| uvp5_sn207_2018_exports_np_sr1812_filtered | 230 | 134 | 2018-08-11 to 2018-09-09 | 51.04 to 49.43 | -131.54 to -145.76 | A.M.P. McDonnell |
| uvp5_sn207_2018_s04p_filtered | 150 | 111 | 2018-03-13 to 2018-05-09 | -59.06 to -75.29 | 179.42 to -179.29 | A.M.P. McDonnell |
| uvp5hd_sn207_2019_i06s_tcn322_filtered | 138 | 44 | 2019-04-16 to 2019-05-11 | -33.23 to -68.35 | 31.53 to 28.09 | A.M.P. McDonnell |
| uvp5_sn210_2018_msm080 | 270 | 127 | 2018-12-27 to 2019-01-25 | -8.5 to -16.4 | -74.17 to -81.0 | R Kiko, H Hauss |
| uvp5_sn210_2019_m156 | 271 | 57 | 2019-07-04 to 2019-07-29 | 21.44 to 17.58 | -16.39 to -24.33 | R Kiko, H Hauss |
| uvp5_sn210_2019_m157 | 272 | 24 | 2019-08-21 to 2019-09-13 | -17.26 to -25.0 | 14.56 to 11.07 | R Kiko, H Hauss |
| uvp5_sn210_2019_m159 | 256 | 46 | 2019-11-02 to 2019-11-18 | 17.6 to -11.5 | -24.25 to -35.02 | R Kiko, H Hauss |
| uvp5_sn210_2019_m160 | 275 | 71 | 2019-11-24 to 2019-12-17 | 18.6 to 14.27 | -19.7 to -25.99 | R Kiko, H Hauss |





| | | | | | | |
|---|---|---|---|---|---|---|
| uvp5_sn210_2020_msm089 | 273 | 46 | 2020-01-18 to 2020-02-16 | 14.03 to 7.25 | -50.83 to -60.08 | R Kiko, H Hauss |
| uvp5_sn221_algoa_bay_2020 | 268 | 36 | 2020-10-28 to 2020-11-23 | -33.73 to -34.03 | 26.29 to 25.7 | Margaux Noyon |





**Table 2**

Table 2: References for datasets published before the revision of the inter-calibration procedure.

| UVP project name | Link to previously published UVP particle dataset |
|---|---|
| uvp5_sn000_tara2009 | https://doi.pangaea.de/10.1594/PANGAEA.836321 |
| uvp5_sn000_tara2010 | https://doi.pangaea.de/10.1594/PANGAEA.836321 |
| uvp5_sn000_tara2011 | https://doi.pangaea.de/10.1594/PANGAEA.836321 |
| uvp5_sn000_tara2012 | https://doi.pangaea.de/10.1594/PANGAEA.836321 |
| uvp5_sn003_tara2013 | https://doi.pangaea.de/10.1594/PANGAEA.836321 |
| uvp5_sn003zp_tara2012 | https://doi.pangaea.de/10.1594/PANGAEA.836321 |
| uvp5_sn001_2012_msm22 | https://doi.org/10.1594/PANGAEA.874871 |
| uvp5_sn001_2012_msm23 | https://doi.pangaea.de/10.1594/PANGAEA.846229 |
| uvp5_sn001_2013_m92 | https://doi.org/10.1594/PANGAEA.885756 |
| uvp5_sn001_2013_m96 | https://doi.pangaea.de/10.1594/PANGAEA.846153 |
| uvp5_sn010_2014_m106 | https://doi.org/10.1594/PANGAEA.874870 |
| uvp5_sn010_2014_m107 | https://doi.org/10.1594/PANGAEA.885759 |
| uvp5_sn010_2015_m119 | https://doi.org/10.1594/PANGAEA.874872 |
| uvp5_sn003_cassiopee_2015 | https://doi.org/10.1594/PANGAEA.876216 |
| uvp5_sn009_2015_p16n | https://doi.org/10.1594/PANGAEA.874875 |
| uvp5_sn202_ps99_21_06_3_filtered | https://doi.pangaea.de/10.1594/PANGAEA.896047 |

515

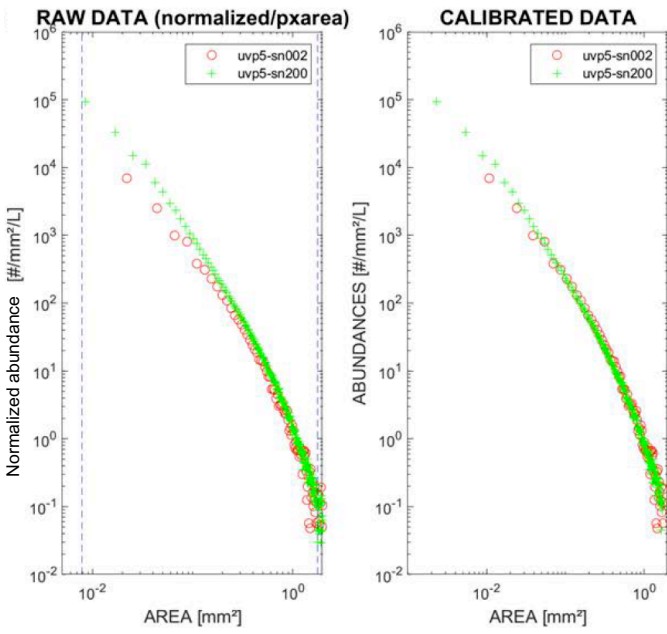

**Figure 1.** UVP5 inter-calibration procedure based on the normalized size spectrum. To calculate the normalized size spectrum, the abundance of particles in a given size class is divided by the mean area of the size class. Normalized abundance of each size class is then plotted against the area of the size class. Figure 1a shows the raw number size spectrum data of the unit to be adjusted (sn200) for one exemplary inter-calibration experiment against sn002 and Fig 1b the respective data after adjustment of the parameters Aa and Exp to coincide better with the number size spectrum of UVP5 sn002.

**Figures**





**Figure 2.** Global distribution of UVP5 data. Lower left panel shows the data distribution in the Mediterranean Sea.



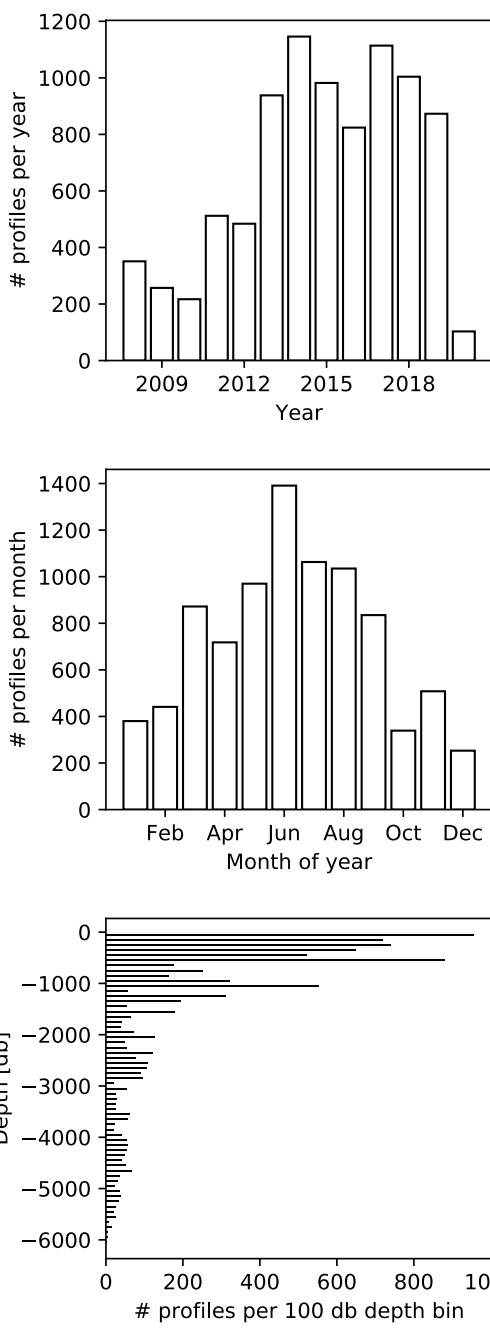

**Figure 3.** UVP5 data distribution per year, month and maximum profile depth (aggregated in 100 dbar depth bins).

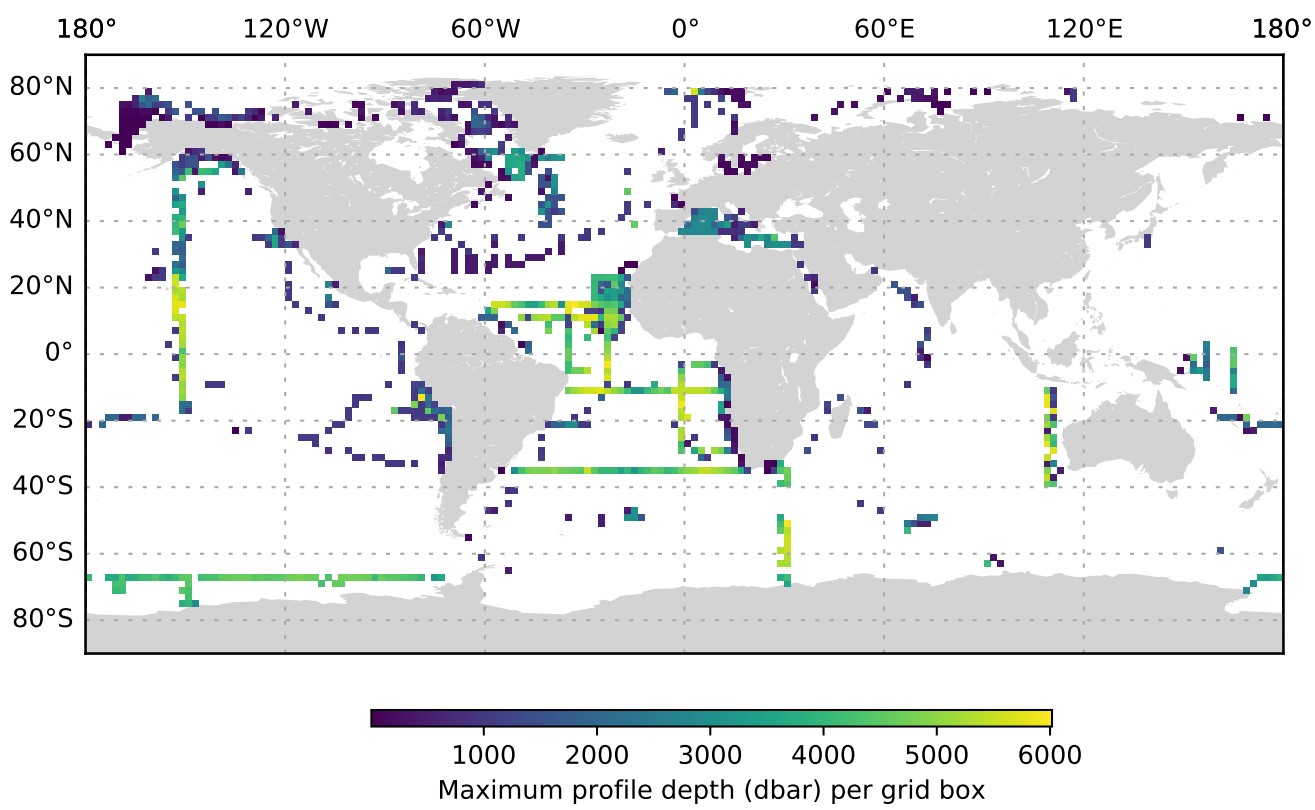

**Figure 4.** Maximum UVP5 profile depth per two degree grid box.

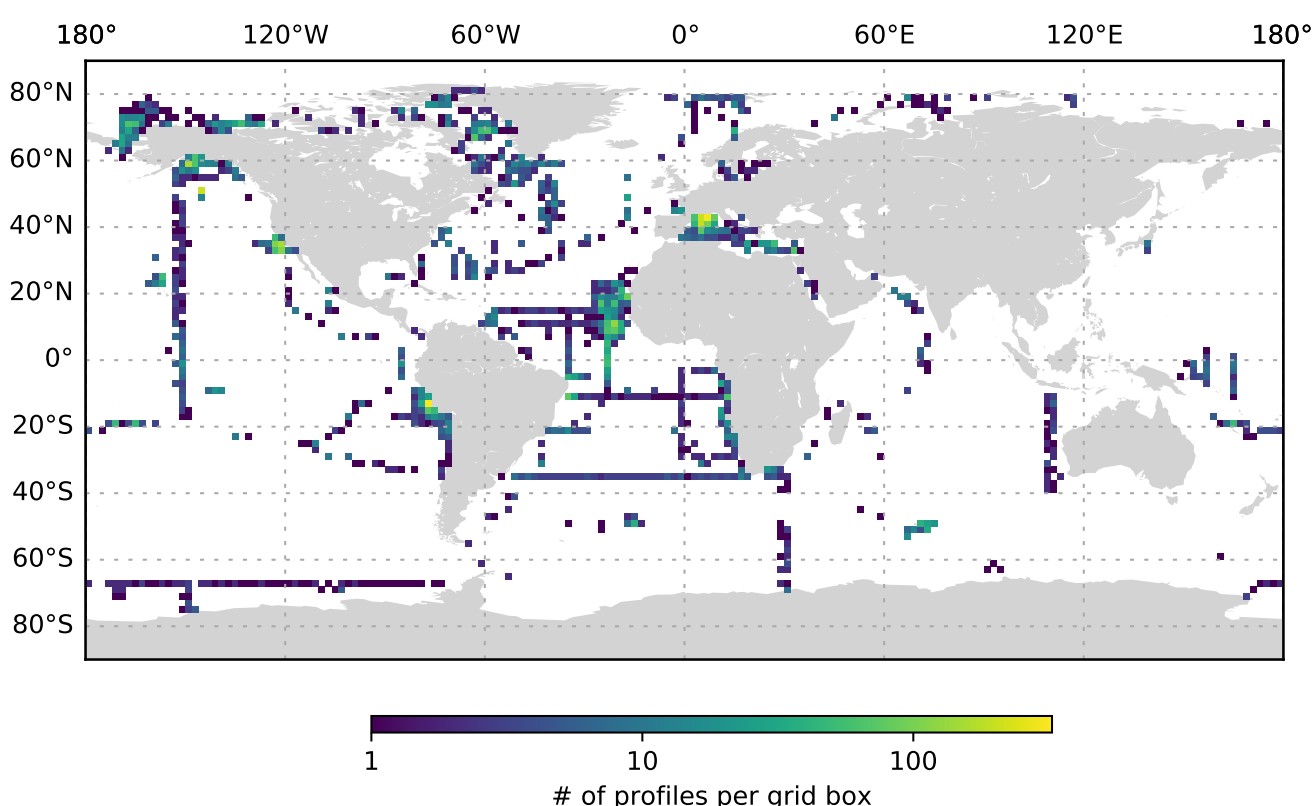

**Figure 5.** UVP5 data distribution per two degree grid box.





**Figure 6.**

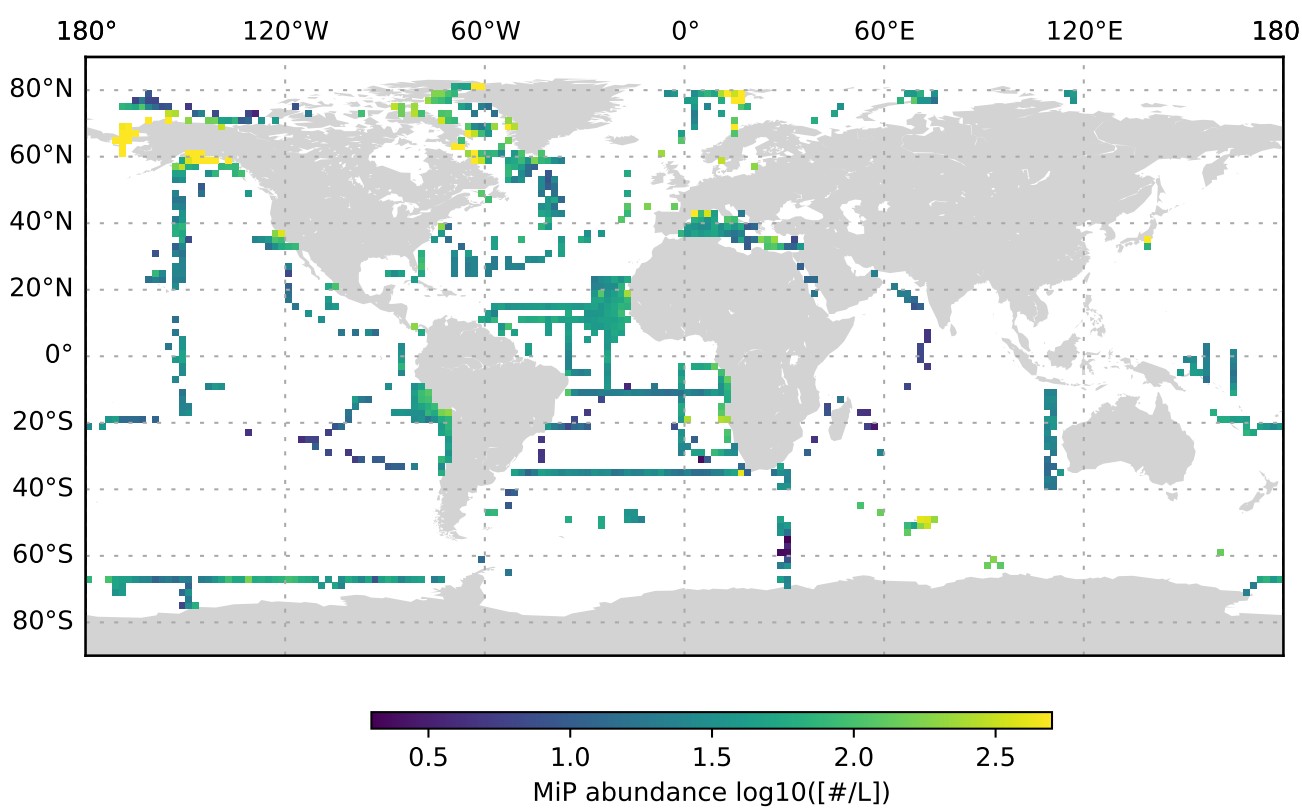

MiP abundance (decadic logarithm) averaged for the 0 to 200 dbar depth layer and per 2 degree grid box. Only profiles at least 200 dbar deep were used for the analysis.



**Figure 7.**

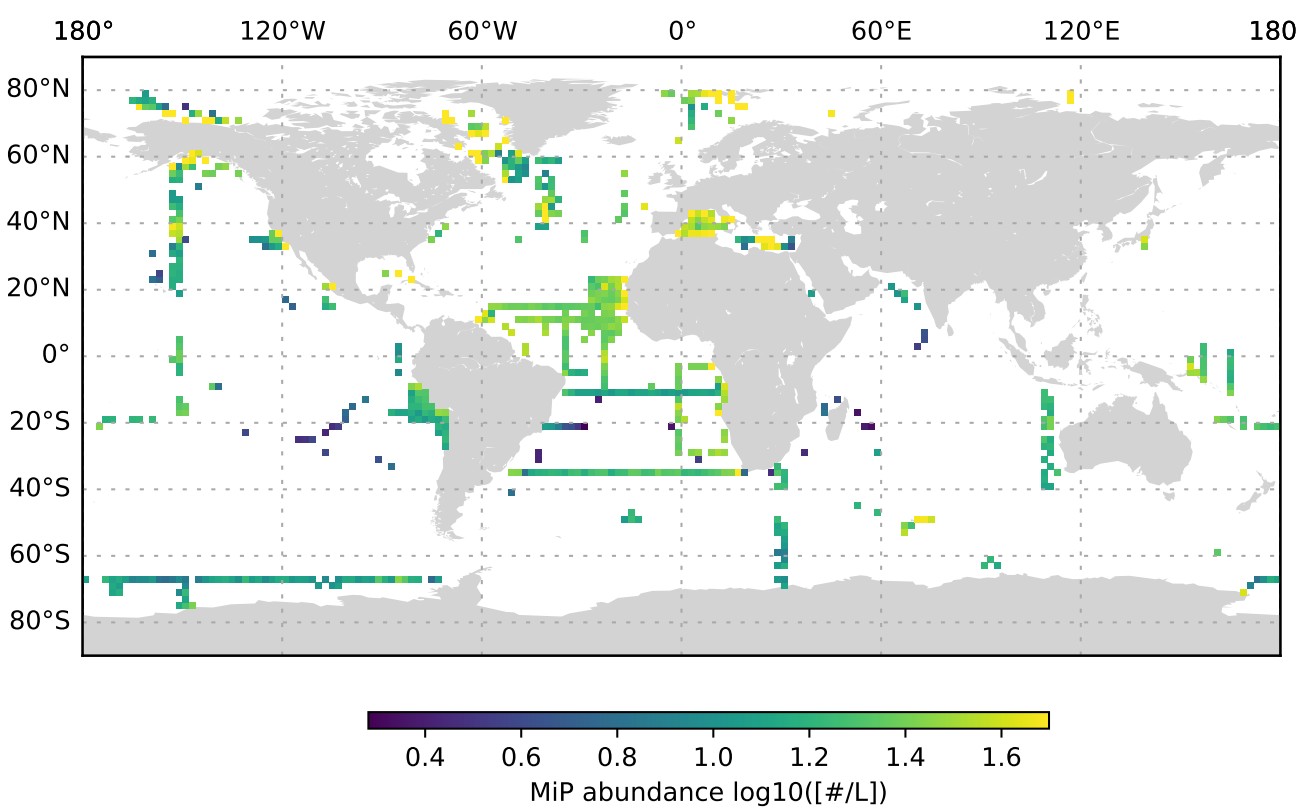

MiP abundance (decadic logarithm) averaged for the 200 to 1000 dbar depth layer and per 2 degree grid box. Only profiles at least 1000 dbar deep were used for the analysis.





**Figure 8.**

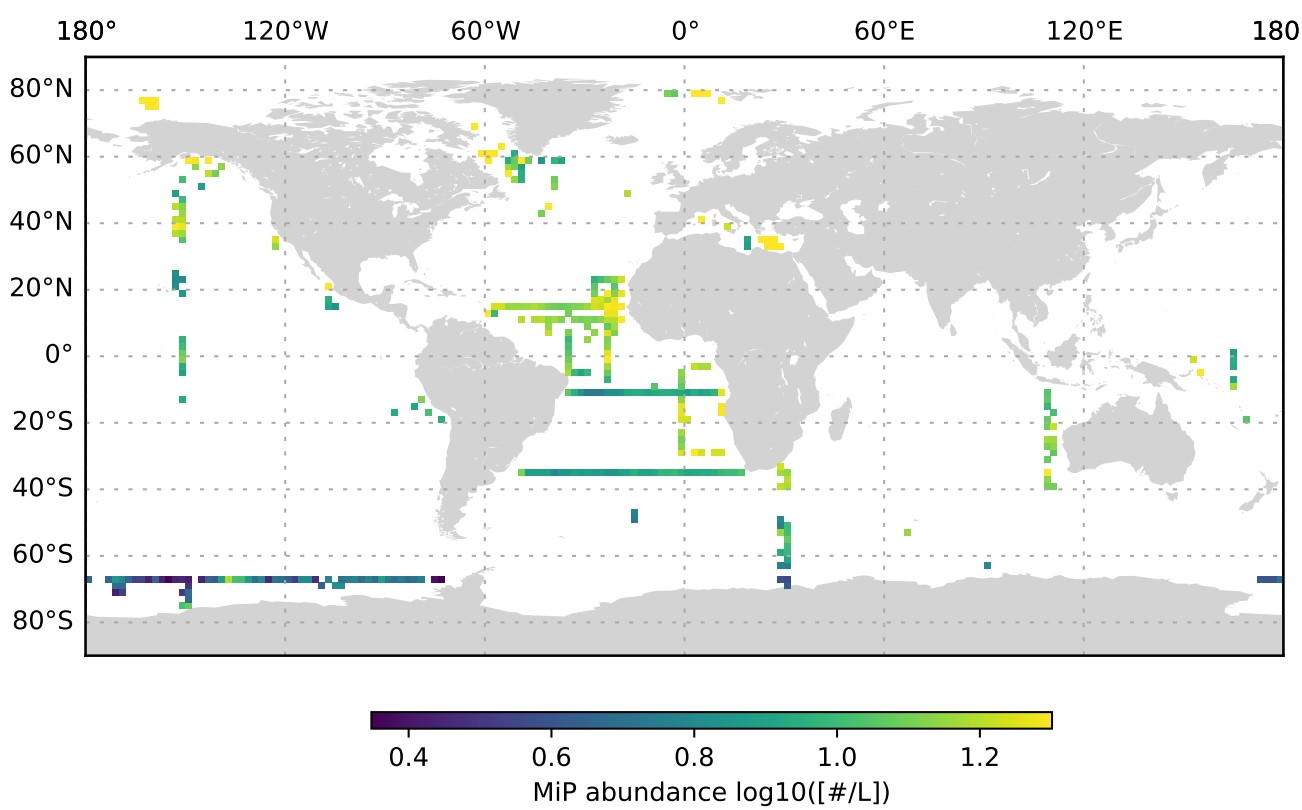

MiP abundance (decadic logarithm) averaged for the 1000 to 3000 dbar depth layer and per 2 degree grid box. Only profiles at least 3000 dbar deep were used for the analysis.

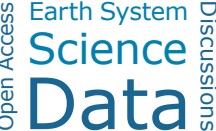

**Figure 9.**

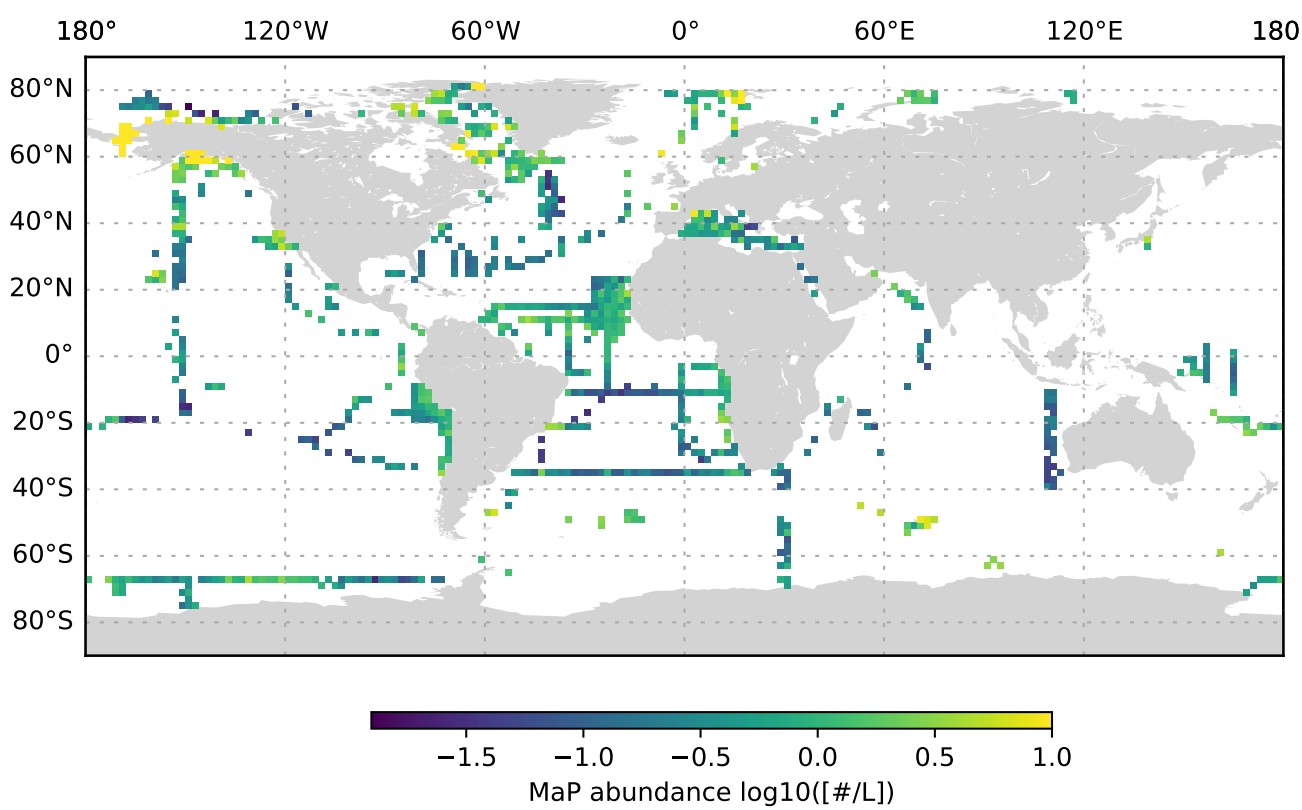

MaP abundance (decadic logarithm) averaged for the 0 to 200 dbar depth layer and per 2 degree grid box. Only profiles at least 200 dbar deep were used for the analysis.



**Figure 10.**

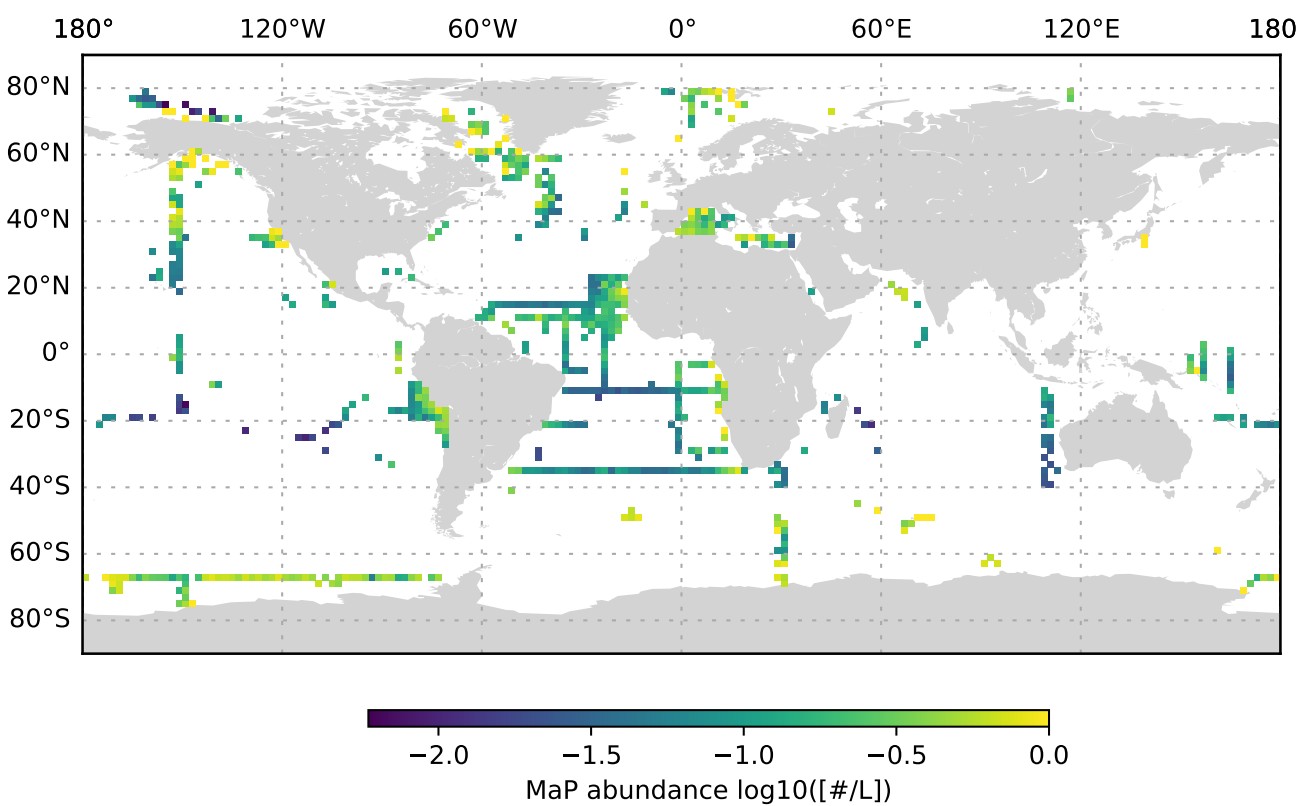

MaP abundance (decadic logarithm) averaged for the 200 to 1000 dbar depth layer and per 2 degree grid box. Only profiles at least 1000 dbar deep were used for the analysis.



**Figure 11.**

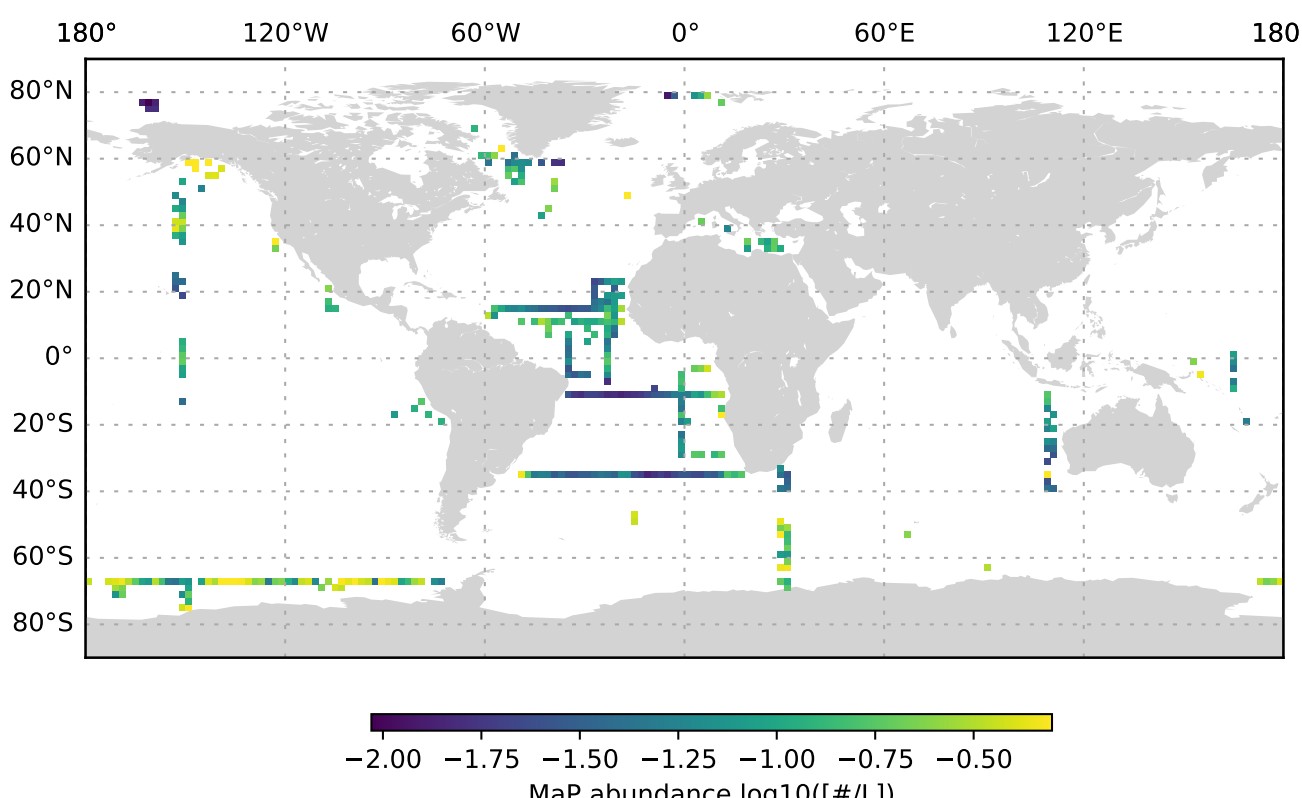

MaP abundance (decadic logarithm) averaged for the 1000 to 3000 dbar depth layer and per 2 degree grid box. Only profiles at least 3000 dbar deep were used for the analysis.





**Figure 12.**

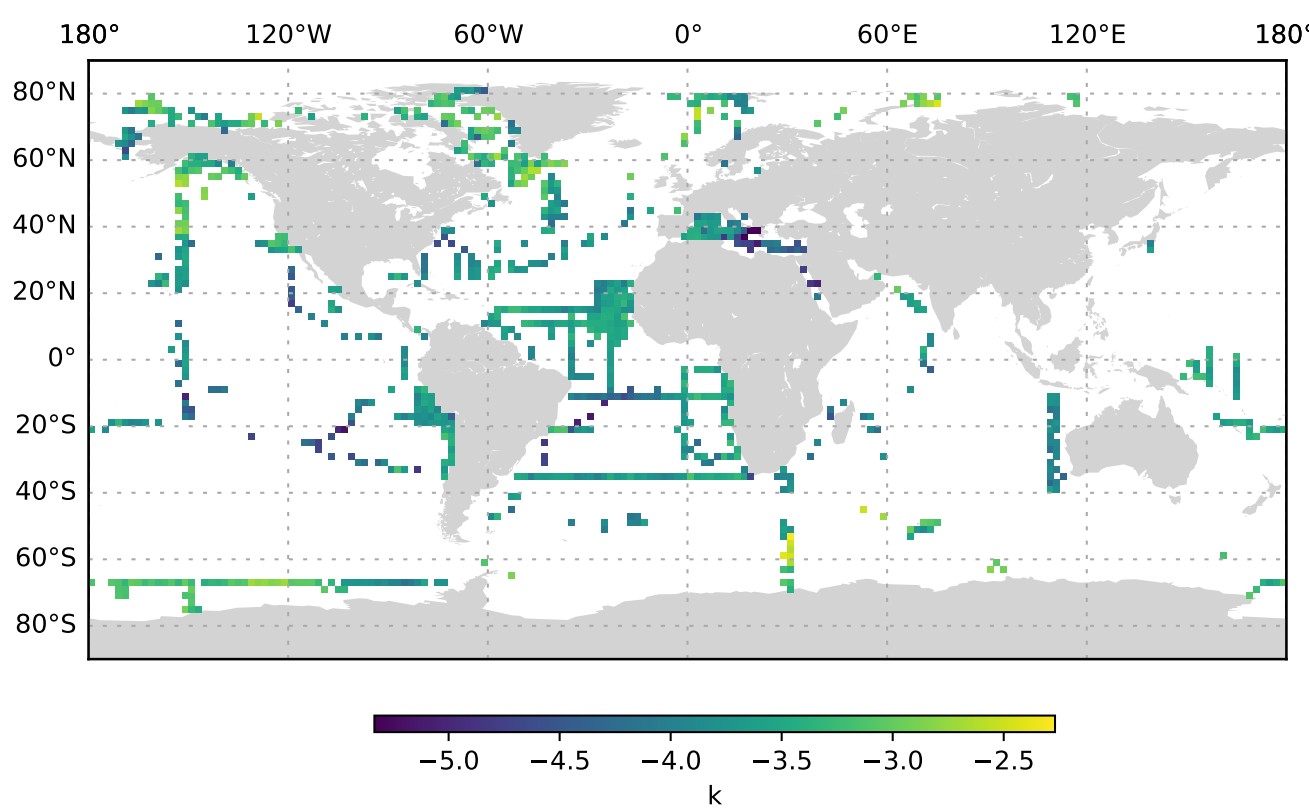

k averaged for the 0 to 200 dbar depth layer and per 2 degree grid box. Only profiles at least 200 dbar deep were used for the analysis.



**Figure 13.**

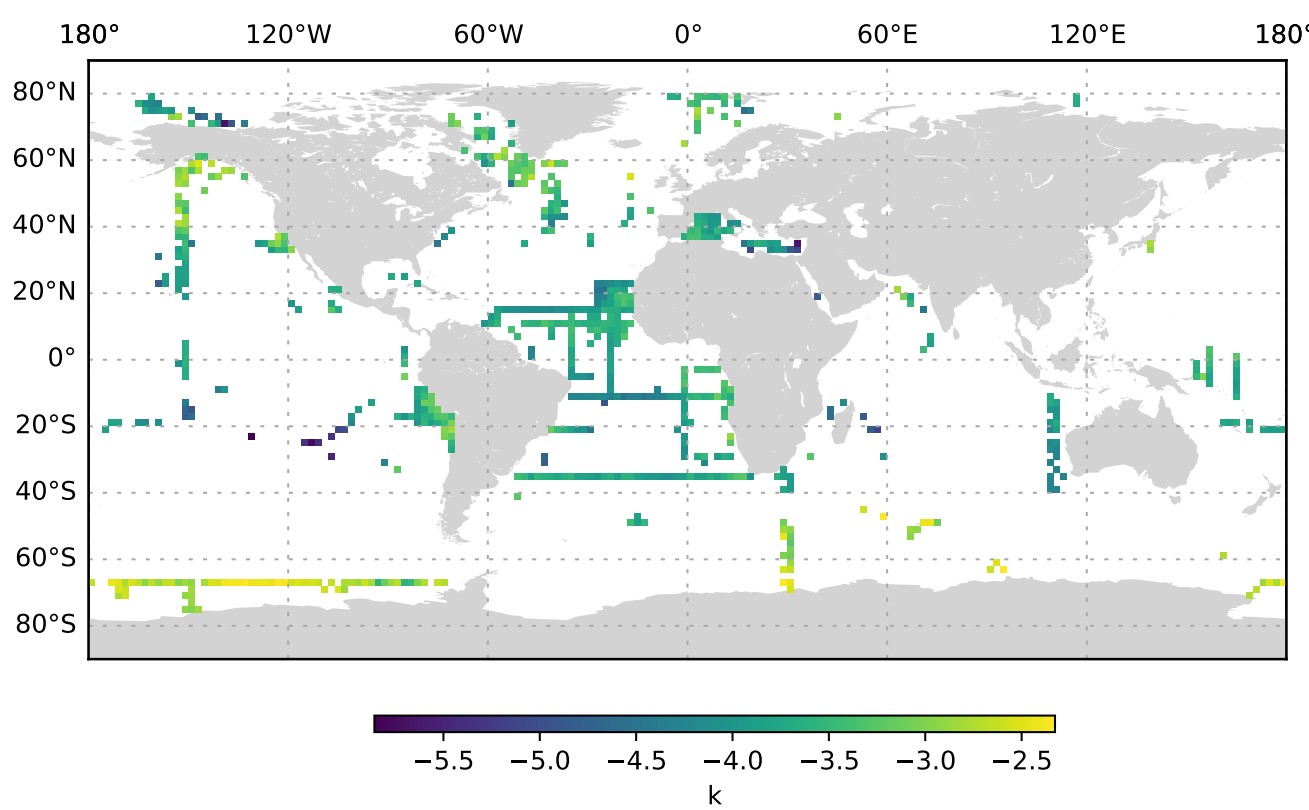

k averaged for the 200 to 1000 dbar depth layer and per 2 degree grid box. Only profiles at least 1000 dbar deep were used for the analysis.



**Figure 14.**

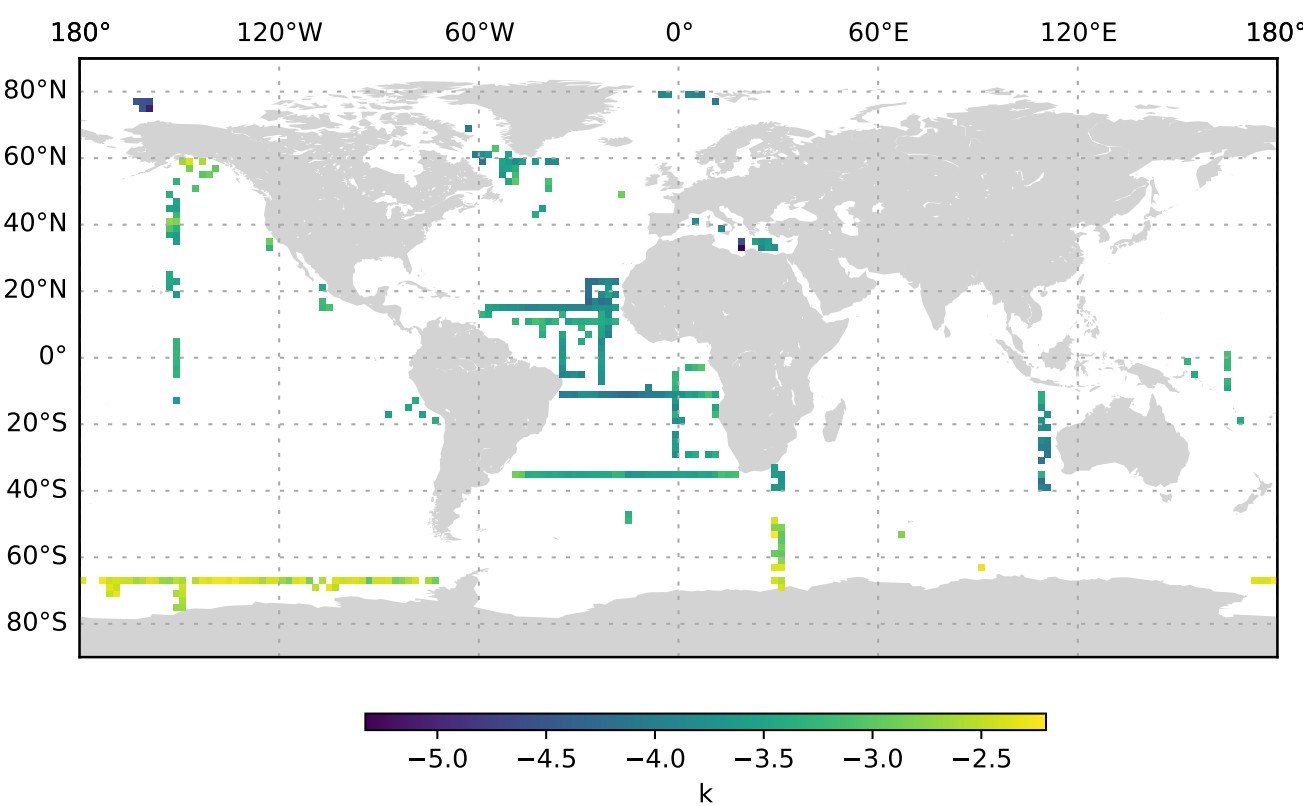

k averaged for the 1000 to 3000 dbar depth layer and per 2 degree grid box. Only profiles at least 3000 dbar deep were used for the analysis.