# Peer review of "A global marine particle size distribution dataset obtained with the Underwater Vision Profiler 5"

_Earth System Science Data, 2022_

## Author Response (AR1)

Dear Referees,

Thank you very much for taking the time to review our manuscript. We have carefully taken your criticism into account and below provide answers to all issues raised in our final author response. We have copied your entire reviews and provide responses where appropriate. We hope that we could satisfactorily resolve all raised issues.

With kind regards,

Rainer Kiko on behalf of all Co-authors.

**Reviewer 1 (https://doi.org/10.5194/essd-2022-51-RC1)**:

The manuscript by Rainer Kiko and his cohort introduces the data base that they have assembled for particle size profiles made around the world and that they placed online to make the database generally available. The data were obtained using UVP5 particle sizers after the instruments had been being calibrated in a consistent manner. The data have been stored in an easily accessible manner using a standard form. Their efforts have produced a valuable resource for the community.The UVP5 is probably the best calibrated optical particle counter used in oceanography. As with all techniques, there are quirks to the data and how they are interpreted that may not be realized by a user of this data set. I believe that it would help new users if the authors added a couple of paragraphs discussing some of the limitations of the data.

The observations are sorted into depth and size ranges, with the number of particles in each depth and diameter range divided by the associated water volume and diameter range to calculate a size spectrum. These are further processed by multiplying the number spectrum by the volume of a particle in the relevant size bin to yield a volume spectrum. The results in the data base are given as these two spectra, which are probably the most useful forms for most people studying particle distributions and dynamics.

Problems with calculated spectral data arise when there is only 1 particle observed in a depth and size range. The number of particles sampled in each depth and size interval can be calculated from the sample volumes and size ranges given in the data tables. Would the sampled water have only one particle of the observed size in any water sample of the same volume? Or, is the observed particle the one lucky enough to be sampled when the average particle concentration is actually one tenth or one hundredth of the above estimate? Because large particles are rarer than small ones, this uncertainty is more of an issue with them. It is particularly a problem for measures that multiply by particle volume, which is greater for the larger particles.

One solution for an individual using the data is to decrease the uncertainty by setting to 0 the spectral value calculated with only one particle. Alternatively, increasing the sample volume by increasing the depth range for each sample would decrease the depth resolution of the observations but should yield more particles in each depth bin, hopefully decreasing the number of size intervals with only 1 particle.

**Authors reply:** We have included the following statement in the section "Data collection, processing, quality control and dataset description" to identify this problem and to highlight the options available to deal with this (lines 170 to 178): "To yield better count statistics at the upper end of the size distribution, UVP users often also combine abundance or biovolume estimates from several depth bins. This is not only possible for the abundance estimates, but also for the biovolume estimates, as the reported biovolumes are the sum of the individual spherical volumes computed from each particle. The aggregation of depth bins leads to a loss of depth resolution, but increases the reliability of the count statistics, especially at the upper end of the size spectrum where particles are rare. To further minimize the uncertainties with singular counts of particles within size classes at the upper limit of the size spectra, one could also set the abundance and biovolume estimate to nan (not a number) if only one particle was observed in the aggregated volume. The count of particles per size bin can be computed by multiplying particle abundance and observed volume. Please also see Bisson et al 2021 for a more in-depth analysis of the impact of count statistics on the estimation of particle abundance and particle flux. Their analysis shows that uncertainties of particle count statistics lead to an approximately 2-fold uncertainty of resulting particle flux estimates."

**Reviewer 1:** Those sampling at sea have the option of simply increasing their sample volumes within a given depth range.

**Authors reply:** We have included the following statement in the section "Recommendations for further instrument usage and growth of the dataset" (lines 302 to 304): "It needs to be reiterated that a larger sampling volume will improve count statistics, especially for larger, rarer particles (Bisson et al 2021). It should therefore be considered in sampling programs to conduct repeated profiles at a station to increase the effective sampled volume for a given station."

**Reviewer 1:** Understanding this issue is important for those who plan to use the data that the authors have accumulated.

**Authors reply:** We thank the reviewer for highlighting this issue and hope that we have better included relevant information in the article.

Smaller issues:

**Reviewer 1:** The manuscript needs to be consistent in its punctuation of references in the text, particularly in the use of parentheses. For example, line 27 uses only one set of parentheses, while line 30 uses two sets. Most of the text has author names inside parentheses for citations; lines 52-55 do not.

**Authors reply:** We have carefully checked the use of parentheses throughout the article. In case of line 30 (and two further cases in the section 'Potential uses of the data', we have kept the use of two parentheses, as the citation is within a parenthesis itself. Latex does not enable the option to provide a citation without parentheses, which in this case would be needed. We hope this problem can be resolved when the paper is set for print. We have fixed the problems in lines 52 to 55.

**Reviewer 1:** In summary, this is an important and useful paper for oceanography. It would benefit from adding couple of paragraphs describing issues associated with using discrete counts as if they are continuous concentrations.

**Authors reply:** As mentioned above, we have provided further information on this topic in the section "Data collection, processing, quality control and dataset description"

**Reviewer 1:** The manuscript also needs another pass to smooth out typographic inconsistencies.

**Authors reply:** The manuscript was carefully checked for typographic inconsistencies.

**Reviewer 2** (https://doi.org/10.5194/essd-2022-51-RC2):

This manuscript describes a very useful dataset and is generally well written. However, there are some ways in which the presentation and dataset could be improved. I divide them into major and minor issues below:

MAJOR ISSUES

**Reviewer 2:** As the authors themselves state, the standard UVP quantifies particles as small as ~102 um and the high-definition version quantifies particles as small as ~64 um (although in my experience even these lower limits should be considered unreliable). However, the authors present data for a size bin starting at ~40um. These lower size bins are HIGHLY unreliable (and likely everything <200-um is questionable). A user who naively downloads the dataset and uses it will not know that these size bins 1) substantially underestimate particle abundance, 2) are very uncertain, because they are derived from a very small number of pixels which can lead to issues with noise, and 3) will have substantially different biases depending on the instrument used (i.e., standard vs. HD, although also some variability instrument-to-instrument within those groups). The authors should consider adding columns that state the confidence associated with each size bin (e.g., a flag that tells users which size bins are reliable) and/or not reporting the data for the size bins <100-um where it is very clear that the dataset is not reliable.

**Authors reply:** We have included a further paragraph in the section "Data collection, processing, quality control and dataset description" to highlight the fact that, due to different instrument settings, different lower size limits of the size spectrum could be realized. We also in this section inform that many users do not use the first or the first two to three size bins in their analyses, as the size measurements here rely on few pixels and therefore might be noisy (lines 157 to 170): "Users of this dataset need to be aware that we provide the data as is, aggregated in 5 dbar depth bins. The particle size that can be quantified reliably with the UVP is limited at the lower end by the optical resolution of the camera and at the upper end by the imaged volume. The optical resolution differs between the different UVP units used. In most cases, the lower size limit is at 0.102 mm (UVP5 SD) or 0.064 mm ESD (UVP5 HD), it is sometimes even lower, i.e., 0.203 or 0.256 mm ESD for early SD deployments. Also, several datasets exist that have a lower limit of 0.0403 mm ESD. In these cases the distance of the camera system to the illuminated field was reduced to increase image resolution. Projects with project ids 33 to 38 (uvp5_sn002zd_cascade2011, uvp5_sn002zd_ccelter_2011, uvp5_sn002zd_gatekeeper2010, uvp5_sn002zd_keops2, uvp5_sn002zd_keops2, uvp5_sn002zd_omer, uvp5_sn002zd_omer_2) and 50 to 51 (uvp5_sn003zp_pelgas2012, uvp5_sn003zp_tara2012) are concerned. In this case the imaged volume was reduced to 0.48 and 0.37 L, respectively. We would like to note that many UVP users do not use the first bin or even the first two or three bins of the size distribution of a given dataset in their analyses, as the particle size estimates at the lower resolution limit only rely on very few image pixels and therefore might be less reliable or noisy The lower limit can be identified by computing a size spectrum with all depth bins of the profile or with the entire project dataset included. The bin where the differential particle size distribution peaks then represents the limit below which data should be considered as not quantitative."

We also point to a preprint (Bisson et al. 2021) where further analyses on the impact of count uncertainties on particle flux estimates are conducted and again caution the user to first carefully analyse the size spectrum of a certain cruise dataset, before using it (lines 177 to 181): "Please also see Bisson et al (2021) for a more in-depth analysis of the impact of count statistics on the estimation of particle abundance and particle flux. Their analysis shows that uncertainties of particle count statistics lead to an approximately 2-fold uncertainty of resulting particle flux estimates. Overall, we recommend careful consideration of the size range to be analysed for each individual cruise or project, as instruments used and their settings differ between each other, which can lead to different count statistics at the lower and upper ends of the size spectrum."

We, however, cannot provide an uncertainty estimate for each size class and each project. This actually depends on particle abundance, imaged volume and instrument setting. We acknowledge that more intensive work on this topic is needed, but also think that the paper by Bisson et al is a first step in this direction.

**Reviewer 1:** This leads to my other concern, which is that the authors provide no uncertainty estimates with their dataset. To be clear, I don't have a great suggestion for this, because it is not clear what approach should be used to accurately determine confidence limits for these measurements. Uncertainty associated with this dataset derives from multiple different factors that are difficult to quantify (sampling volume and number of particles sampled in the size class and depth bin, accuracy of defining particle size due to shape and orientation,

calibration of the instrument, avoidance behavior by swimming organisms, etc.). Recognizing these issues, it is not clear what approach should be taken to define confidence limits. Nevertheless, the utility of the dataset would be increased substantially if the authors could provide reasonable confidence limits (especially since confidence is likely much higher for 200 – 1000 um particles than for either smaller or larger particles/aggregates).

**Authors reply:** We agree with the reviewer that such confidence limits would be beneficial, but also do not see a simple way forward. As mentioned above, we now further highlight the issue that count statistics will be bad at the upper and lower limit of the size spectrum and refer the reader to Bisson et al. 2021, where an assessment is conducted how uncertainties in count statistics propagate into uncertainties of flux estimates.

MINOR ISSUES *(line numbers refer to the manuscript under discussion, not the revised version)*

**Line 11:** It seems redundant to state that 19% were shallower than 200 and 80% were deeper than 200. I would recommend saving words by just stating one or the other

**Reply:** We have removed the statement that 80% of the profiles were deeper than 200 dbar.

**Line 13:** Probably best to modify "particle abundance is found to be high at high latitudes" to "particle abundance is found to be high at high latitudes in the summer", since there is very little data from high latitude winters and it is reasonable to expect that high latitude winters might have different particle loads

**Reply:** We have kept the formulation as is, as we are not discussing seasonality in the abstract.

**Lines 22-24:** It is not clear whether it is living, detrital, or abiotic particles that can vary from 1% to 50%. I would assume that the authors mean living particle fraction can vary from 1 to 50%, because previous studies have shown that detritus can comprise between 50 and 90%.

**Reply:** We have corrected this error by reducing the statement to only the living particles.

**Line 29:** I would delete the hydrothermal vents example, because it seems to imply that that chemoautotrophy is only common in unique environments, when research over the past decade has suggested that it may be widespread in the ocean – and it is the low background amount of chemoautotrophy that is most likely to contribute organic matter to the particles being studied here.

**Reply:** We have deleted the hydrothermal vent example.

**Line 54:** There is an issue with the parentheses in this line.

**Reply:** The issue was resolved

**Line 150:** I appreciate that the authors have included a CTD filename, but is it possible to also provide a pointer to where the CTD data can be found? That seems like it would substantially increase the usability of the dataset

**Reply:** This information is provided in the metafile that is also provided at Pangaea, together with the dataset. We have now included the information that the links to CTD data are included in the metadata file (lines 154 to 156).

**Lines 171 – 172:** a 4.2 and 1.5X factor increase in abundance based on a re-calibration is huge. The authors should briefly discuss what led to such a large change.

**Reply:** We have added the following text to qualitatively explain these factors (lines 200 to 202): In our view, these changes are related to the increased resolution of the HD version (compared to the SD version) that enabled us to better quantify small particles during the initial laboratory calibration experiment for UVP5 HD sn203. This improved calibration was then propagated to all other units and superseded the earlier calibration experiment done with an SD unit.

**Line 206:** Seems to be a typo at the beginning of this line.

**Reply:** We can not spot a typo here.

**Lines 217 – 219:** Considering the huge ranges here and the fact that the arithmetic mean will be heavily weighted towards high measurements, I would recommend also reporting the median.

**Reply:** We have added median values for all estimates.

**Line 221:** "interpreted as a consequence of microbial and metazoan flux attenuation". While this isn't necessarily an incorrect statement, the real reason that there are more particles near the surface is simply that the surface is the region of photosynthesis and hence most particle production.

**Reply:** We agree that this statement is not wrong and therefore kept it as is. Actually, without microbial and metazoan flux attenuation, one could well imagine a situation in which there are more particles at depth than at the surface. This could e.g. be the case if particles were produced at the surface at a certain rate, sank to depth and then accumulated at depth e.g. as density increases with depth or if they disaggregated and thereby sank more slowly.

**Reviewer 2:** The authors have a tendency to cite UVP studies even for citations that are not specifically UVP related (for instance, they cite Stemmann et al. 2004 and Guidi et al. 2009 for the basic concept of microbial and metazoan flux attenuation, but these concepts were understood well before those papers).

**Reply:** We agree with the reviewer that we focus throughout this paper on the use of UVP

data to reveal ecological and biogeochemical patterns and processes. We did not intend to write a review about any of these topics and think that we have provided a fair overview of the general literature dealing with particle distribution in the ocean in the introduction. We therefore decided to keep the citation pattern as is.

**Reviewer 2:** This might be personal preference, but I do not like the colormap used in the figures. My eyes can really only differentiate individual points into relatively high or relatively low particle abundance. Other color maps allow a more precise visualization of the data.

**Reply:** The colormaps used is viridis, which is one of the perceptually uniform colormaps available for python (https://matplotlib.org/stable/tutorials/colors/colormaps.html). We feel that this is a good choice and would like to keep this colormap. Please also see e.g. https://stats.stackexchange.com/questions/223315/why-use-colormap-viridis-over-jet regarding the choice of colormaps.

---

## Author Response (AR2)

Response to editorial question:

Regarding the figure #1: Please ensure that the colour schemes used in your maps and charts allow readers with colour vision deficiencies to correctly interpret your findings. Please check your figures using the Coblis – Color Blindness Simulator (https://www.color-blindness.com/coblis-color-blindness-simulator/) and revise the colour schemes accordingly.

Response: We checked figure #1 at the Color Blindness Simulator. As not only different colours (red, green), but also different symbols (circle, cross) are used to depict the two different data types shown in figure 1, it is readable for every types of Color Blindness

Please note that I somehow could not upload the main.tex file, the files containing the data for table creation and the figure files. The main.tex file was somehow not accepted and the plots.zip folder was rejected for holding further folders, although it did not. There was no option to upload the table data files.

I have uploaded these files here as main.tex, plots.zip and tables.zip:

https://we.tl/t-7hIZxNVn3h